# DENOISING TASK ROUTING FOR DIFFUSION MODELS

**Byeongjun Park**[♠][†]  **Sangmin Woo**[♠][†]  **Hyojun Go**[♡][†]  **Jin-Young Kim**[♡][†]  **Changick Kim**[♠][*]

[♠]KAIST       [♡]Twelve Labs       (†: Equal contribution,     ∗: Corresponding author)

{pbj3810, smwoo95, changick}@kaist.ac.kr
{gohyojun15, seago0828}@gmail.com

## ABSTRACT

Diffusion models generate highly realistic images by learning a multi-step denoising process, naturally embodying the principles of multi-task learning (MTL). Despite the inherent connection between diffusion models and MTL, there remains an unexplored area in designing neural architectures that explicitly incorporate MTL into the framework of diffusion models. In this paper, we present Denoising Task Routing (DTR), a simple add-on strategy for existing diffusion model architectures to establish distinct information pathways for individual tasks within a single architecture by selectively activating subsets of channels in the model. What makes DTR particularly compelling is its seamless integration of prior knowledge of denoising tasks into the framework: (1) Task Affinity: DTR activates similar channels for tasks at adjacent timesteps and shifts activated channels as sliding windows through timesteps, capitalizing on the inherent strong affinity between tasks at adjacent timesteps. (2) Task Weights: During the early stages (higher timesteps) of the denoising process, DTR assigns a greater number of task-specific channels, leveraging the insight that diffusion models prioritize reconstructing global structure and perceptually rich contents in earlier stages, and focus on simple noise removal in later stages. Our experiments reveal that DTR not only consistently boosts diffusion models' performance across different evaluation protocols without adding extra parameters but also accelerates training convergence. Finally, we show the complementarity between our architectural approach and existing MTL optimization techniques, providing a more complete view of MTL in the context of diffusion training. Significantly, by leveraging this complementarity, we attain matched performance of DiT-XL using the smaller DiT-L with a reduction in training iterations from 7M to 2M. Our project page is available at https://byeongjun-park.github.io/DTR/.

## 1 INTRODUCTION

Diffusion models (Ho et al., 2020; Sohl-Dickstein et al., 2015; Song et al., 2021) have made significant strides in generative modeling across various domains, including image (Dhariwal & Nichol, 2021; Rombach et al., 2022), video (Harvey et al., 2022), 3D (Woo et al., 2023), audio (Kong et al., 2020) and natural language (Li et al., 2022). In particular, they have demonstrated their versatility across a broad spectrum of image generation scenarios such as unconditional (Ho et al., 2020; Song et al., 2020), class-conditional (Dhariwal & Nichol, 2021; Nichol & Dhariwal, 2021), and text-to-image generation (Nichol et al., 2021; Ramesh et al., 2022; Saharia et al., 2022).

Diffusion models are designed to learn denoising tasks across various noise levels by reversing the forward process that distorts data towards a predefined noise distribution. Recent studies (Hang et al., 2023; Go et al., 2023a) have shed light on the multi-task learning (MTL) (Caruana, 1997) aspect inherent in diffusion models, where a single neural network handles multiple denoising tasks. They particularly focus on enhancing the optimization of MTL in diffusion models, employing techniques such as loss weighting (Hang et al., 2023) and task clustering (Go et al., 2023a), aiming to address the issue of *negative transfer* — a phenomenon that arises when shared parameters are between conflicting tasks. While these efforts demonstrate the promise of viewing diffusion models as MTL, there remains an unexplored avenue for designing neural architectures from an MTL perspective within the context of diffusion models.

One common practice in diffusion models is to condition the models with timesteps (or noise levels) through differentiable operation (Ho et al., 2020; Karras et al., 2022; Rombach et al., 2022), prompting the model's behavior by adjusting representation of model according to timesteps (or noise levels). This can be seen as an *implicit* way of incorporating MTL aspects into the architectural design. However, we argue that this may not fully address negative transfer, as it places the entire burden of task adaptability solely on *implicit* signals.

In this paper, we take a step beyond implicit conditioning and *explicitly* tackle multiple denoising tasks by making a simple modification to existing diffusion model architectures. Specifically, we draw inspiration from prior works on task routing (Strezoski et al., 2019; Ding et al., 2023), which enables the establishment of distinct information pathways for individual tasks within a single model. The distinct information pathways are implemented through task-specific channel masking, making task routing effectively handle numerous tasks (Strezoski et al., 2019). However, we observe that a naive random routing approach (Strezoski et al., 2019), which allocates random pathways for each task, does not take account into the inter-task relationship between denoising tasks in diffusion models, resulting in a detrimental impact on performance.

To tackle this challenge, we present the Denoising Task Routing (DTR), a simple add-on strategy for existing diffusion model architectures. DTR enhances them by establishing task-specific pathways that integrate prior knowledge of diffusion-denoising tasks, such as: **(1) Task Affinity:** Considering strong task affinity between adjacent timesteps (Go et al., 2023a), DTR activates similar channels for tasks at adjacent timesteps by sliding windows over channels throughout the timesteps and activating channels within the window. **(2) Task Weights:** Inspired by the observation that diffusion models prioritize reconstructing the global structure and perceptually rich contents in the early stages (higher timesteps) (Choi et al., 2022), DTR allocates an increased number of task-specific channels to denoising tasks at higher timesteps.

Building upon this foundation, DTR offers notable advantages: **(1) Simple Implementation**: DTR can be integrated with minimal lines of code, streamlining its adoption. **(2) Elevated Performance**: DTR significantly elevates the quality of the generated samples. **(3) Accelerated Convergence**: DTR enhances the convergence speed of existing diffusion models. **(4) Efficiency**: DTR achieves these without extra parameters and incurs only a negligible computational cost for channel masking.

Finally, we conduct experiments across various image generation tasks, such as unconditional, class-conditional, and text-to-image generation, with FFHQ (Karras et al., 2019), ImageNet (Deng et al., 2009), and MS-COCO dataset (Lin et al., 2014), respectively. By incorporating our proposed DTR into two prominent architectures, DiT (Peebles & Xie, 2022) and ADM (Dhariwal & Nichol, 2021), we observe a significant enhancement in the quality of generated images, thereby validating the benefits of our DTR. Moreover, we demonstrate the seamless compatibility of MTL optimization techniques tailored for diffusion models with our MTL architectural design for DTR. Significantly, we attain similar DiT-XL's performance using the smaller DiT-L with a reduction in training iterations from 7M to 2M, showcasing the efficiency and effectiveness of our approach.

## 2 RELATED WORK

**Diffusion model architecture.** Advancements in diffusion model architecture center on integrating well-established architectural components into the framework of diffusion models. Earlier works use a UNet-based architecture (Ronneberger et al., 2015) and propose several improvements. For example, DDPM (Ho et al., 2020) uses group normalization (Wu & He, 2018) and self-attention (Vaswani et al., 2017), IDDPM (Nichol & Dhariwal, 2021) uses multi-head self-attention, Song et al. (2021) proposes to scale skip connections by $1/\sqrt{2}$, and ADM (Dhariwal & Nichol, 2021) proposes the adaptive group normalization. Recently, several works propose transformer-based architectures for diffusion models instead of UNet, including GenViT (Yang et al., 2022), U-ViT (Bao et al., 2023), RIN (Jabri et al., 2022), DiT (Peebles & Xie, 2022) and MDT (Gao et al., 2023). Unlike these works, our objective is to incorporate the MTL aspects into architectural design. Specifically, we propose a simple add-on strategy to improve existing diffusion models with task routing, and we validate our method upon both representative UNet and Transformer-based architectures, ADM and DiT.

**Multi-task learning (MTL).** MTL (Caruana, 1997; Sener & Koltun, 2018) aims to improve efficiency and prediction accuracy across multiple tasks by sharing parameters and learning them simultaneously. This approach stands in contrast to training separate models for each task, allowing

the model to leverage inductive knowledge transfer among related tasks. However, MTL encounters challenges that conflicting tasks exist, leading to a phenomenon known as *negative transfer* (Ruder, 2017), where knowledge learned in one task negatively impacts the performance of another.

To address the negative transfer, previous research explores optimization strategies and architectural designs. Optimization strategies focus on mitigating two main problems: (1) conflicting gradients and (2) unbalanced losses or gradients. Conflicting gradients between tasks can cancel each other thus resulting in suboptimal updates. To mitigate this, Yu et al. (2020) project a gradient onto the normal plane of conflicting gradient and Chen et al. (2020) stochastically drop elements in gradients. Imbalanced learning, where tasks with larger losses or gradients dominate the training, is also addressed through loss balancing (Kendall et al., 2018) and gradient balancing (Navon et al., 2022).

In terms of MTL architectures, researchers develop both implicit and explicit methods. Implicit methods guide the model to learn multiple tasks with task embeddings, avoiding the extensive task-specific modifications (Sun et al., 2021; Zhang et al., 2018; Popovic et al., 2021; Pilault et al., 2021). Explicit methods, on the other hand, embed task-specific behaviors directly into the architecture through task-specific branches (Long et al., 2017; Vandenhende et al., 2019), task-specific modules (Liu et al., 2019; Maninis et al., 2019), feature fusion across multiple network branches (Gao et al., 2019; Misra et al., 2016), and task routing mechanisms (Strezoski et al., 2019; Pfeiffer et al., 2023). Task routing, in particular, demonstrates its scalability while requiring minimal additional parameters, making it suitable for handling a large number of tasks. Therefore, in our work, we adopt task routing to enhance explicit MTL design within existing diffusion model architectures. Furthermore, in contrast to prior research on task routing (Strezoski et al., 2019; Pfeiffer et al., 2023; Pascal et al., 2021), it is noteworthy that our proposed method introduces a novel approach that incorporates priors for inter-task relationships without the need for extra parameters.

**MTL contexts in diffusion models.** Recent studies (Hang et al., 2023; Go et al., 2023a) revisit diffusion models as a form of MTL, where a single neural network simultaneously learns multiple denoising tasks with various noise levels. They observe negative transfer between denoising tasks and seek to enhance diffusion models by addressing the issue from an MTL optimization perspective. However, there remains limited exploration of architectural improvements from an MTL architectural perspective. To bridge this gap, our work proposes architectural enhancements within the framework of MTL for diffusion models.

Conditioning the model with timestep (Ho et al., 2020) or noise level (Song et al., 2021) can be perceived as an *implicit* method of incorporating MTL aspects into architectural design. For instance, DDPM (Ho et al., 2020) adds the Transformer sinusoidal position embedding (Vaswani et al., 2017) into each residual block, which is widely adopted for various diffusion models including LDM (Rombach et al., 2022), ADM (Dhariwal & Nichol, 2021), DiT (Peebles & Xie, 2022) and EDM (Karras et al., 2022). However, we argue that relying solely on implicit signals is insufficient for effectively mitigating negative transfer. Our goal in this paper is to explicitly incorporate prior knowledge of denoising tasks into the existing diffusion model architectures with task routing.

## 3 PRELIMINARY

**Diffusion models.** Diffusion models (Dhariwal & Nichol, 2021; Song et al., 2020) stochastically transform an original data $\boldsymbol{x}_0$ into latent, often following a Gaussian distribution, by iteratively adding noise — the *forward process*. To make diffusion models generative, they need to learn to reverse the perturbed data back to its original distribution $p(\boldsymbol{x}_0)$ — the *reverse process*. The forward process can be conceptualized as a fixed-length Markov chain comprising $T$ discrete steps. At each timestep $t$ along this chain, represented as $\boldsymbol{x}_t$, the data undergoes a transformation based on a conditional distribution $q(\boldsymbol{x}_{1:T}|\boldsymbol{x}_0)$. Specifically, $q(\boldsymbol{x}_t|\boldsymbol{x}_0)$ is modeled as a Gaussian distribution $\mathcal{N}(\boldsymbol{x}_t; \sqrt{\bar{\alpha}_t}\boldsymbol{x}_0, (1 - \bar{\alpha}_t)\mathbf{I})$, where $\bar{\alpha}_t$ represents a noise schedule parameter, and $\boldsymbol{x}_t$ denotes the noisy version of the input $\boldsymbol{x}_0$ at time $t$. The reverse process recovers the original data by modeling $p(\boldsymbol{x}_{t-1}|\boldsymbol{x}_t)$, which approximates the distribution $q(\boldsymbol{x}_{t-1}|\boldsymbol{x}_t)$. This equips the model to effectively "undo" the diffusion steps and reconstruct the original data from the noisy observations. To achieve this, many diffusion models commonly use the training strategy of DDPM (Ho et al., 2020), which aims to optimize a noise prediction network $\boldsymbol{\epsilon}_{\boldsymbol{\theta}}(\boldsymbol{x}_t, t)$ by minimizing a simple objective $\sum_{t=1}^{T} \mathcal{L}_t$ with respect to $\theta$, where $\mathcal{L}_t$ is defined as:

$$\mathcal{L}_t := \mathbb{E}_{\boldsymbol{x}_0, \boldsymbol{\epsilon} \sim \mathcal{N}(0,1)} \|\boldsymbol{\epsilon} - \boldsymbol{\epsilon}_{\boldsymbol{\theta}}(\boldsymbol{x}_t, t)\|_2^2. \tag{1}$$

**Task routing.** Task routing (Strezoski et al., 2019; Ding et al., 2023) is proposed to explicitly establish task-specific pathways within a single neural network. In practice, task routing employs a $C$-dimensional task-specific binary mask $\boldsymbol{m}_D \in \{0,1\}^C$ associated with the task $D$. Formally, the task routing is implemented by task-specific channel masking at the $l$-th layer, given the input $\boldsymbol{z}^l$ and a transformation function $F^l$, can be expressed as:

$$\boldsymbol{z}^{l+1} = \boldsymbol{m}_D \odot F^l(\boldsymbol{z}^l), \tag{2}$$

where $\odot$ denotes channel-wise multiplication. We note that this operation performs a conditional feature-wise transformation, allowing the neural network to create task-specific subnetworks within a single model. By explicitly separating in-model data flows, the neural network builds its own beneficial sharing practices, effectively addressing negative transfer issues that may arise from sharing channels between conflicting tasks. One significant advantage of task routing lies in its scalability. It does not significantly increase the number of parameters or computational complexity with the addition of tasks. As demonstrated in (Strezoski et al., 2019), task routing exhibits excellent scalability, proving its effectiveness even for scenarios with hundreds of tasks.

## 4 METHODOLOGY

In this section, we introduce Denoising Task Routing (DTR), a straightforward add-on strategy on existing diffusion model architectures to enhance the learning of multiple denoising tasks. We first describe our DTR, focusing on the integration of the task routing framework (Strezoski et al., 2019; Ding et al., 2023) into the diffusion model framework in Sec. 4.1. Next, we consider a naive random routing method and discuss its limitations on handling prior knowledge of denoising tasks in Sec. 4.2. Finally, we present a detailed description of DTR in Sec. 4.3, which explicitly considers the prior knowledge of denoising tasks in the routing mask creation.

### 4.1 TASK ROUTING FOR DIFFUSION MODELS

We conceptualize diffusion training as a form of MTL, where each task corresponds to the denoising task $D_t$ at a specific timestep $t \in \{1, \ldots, T\}$ learned by $\mathcal{L}_t$ in Eq. (1). Typically, $T$ often surpasses 1000, resulting in thousands of denoising tasks being jointly optimized in a single model.

Many diffusion models employ a multi-layered residual block structure in their architecture (Rombach et al., 2022; Dhariwal & Nichol, 2021; Peebles & Xie, 2022; Song et al., 2021). These models commonly adopt the practice of initializing each residual block as the identity function. For example, ADM (Dhariwal & Nichol, 2021) initializes the final convolutional layer of every residual block as zero. On the other hand, DiT-based models (Peebles & Xie, 2022; Gao et al., 2023) utilize adaLN-Zero in their transformer blocks for initializing them as identity functions. To easily integrate these practices into task routing, we apply task routing at the level of residual blocks, emphasizing block-wise task routing as the foundational element of our method. By denoting the $l$-th block as $\mathrm{Block}^l$ and its input as $\boldsymbol{z}^l \in \mathbb{R}^{H \times W \times C}$, our denoising task routing is represented as:

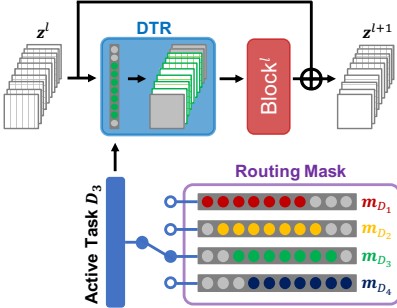

Figure 1: **The overview of DTR.** DTR makes explicit task-specific pathways by channel masking.

$$\boldsymbol{z}^{l+1} = \boldsymbol{z}^l + \mathrm{Block}^l(\boldsymbol{m}_{D_t} \odot \boldsymbol{z}^l), \tag{3}$$

where $\boldsymbol{m}_{D_t} \in \{0,1\}^C$ denotes task-specific binary mask for $D_t$.

Figure 1 provides a concise overview of our DTR scheme, which is adaptable to general residual block structures including ADM (Dhariwal & Nichol, 2021) and DiT (Peebles & Xie, 2022). In both the inference and training stages, the activated mask is set according to the input timestep $t$ of the noise prediction network $\epsilon_\theta$. Through this approach, we explicitly establish task-specific pathways within a single noise prediction network $\epsilon_\theta$. Detailed implementations for incorporating DTR in ADM and DiT architectures can be found in Appendix A.

## 4.2 A NAIVE RANDOM TASK ROUTING APPROACH (R-TR)

The remaining part of designing task routing is to establish task-specific routes by defining task-specific routing mask $\boldsymbol{m}_{D_t}$. To establish task-specific routes, we first consider setting $\boldsymbol{m}_{D_t}$ as random masks (Strezoski et al., 2019), which activates a predefined portion of random channels for each task. For each denoising task $D_t$, we randomly sample $C_\beta = \lfloor \beta C \rfloor$ channel indices from the set $\{1, \ldots, C\}$, where $0 < \beta \leq 1$. Then, the routing mask $\boldsymbol{m}_{D_t}$ is configured to assign a value of one to the randomly sampled channel indices and a value of zero to others.

Here, activation ratio $\beta$ determines the trade-off between task-specific units versus task-general units within the model architecture. When $\beta = 1$, the model is the same as the original model without task routing, where all units are shared among tasks. As $\beta$ decreases from 1, the model allocates fewer units for sharing, thereby enhancing task-specificity.

However, employing random masking for diffusion models might overlook the inter-task relationships between denoising tasks. A recent work (Go et al., 2023a) shows that affinity between denoising tasks increases as the proximity of their timestep increases, suggesting that sharing units between tasks with closer timesteps is beneficial. Despite this, the expected number of shared channels remains constant for pairs of distinct tasks (as we observed in Appendix B), implying that random masking inherently cannot consider timestep proximity. Additionally, we empirically validate this in Sec. 5 and results can be found in Table 1. To leverage the prior knowledge of the diffusion task, we design a more tailored mask in the next section.

## 4.3 MASK CREATION OF DENOISING TASK ROUTING (DTR)

To adequately reflect the specific characteristics of diffusion denoising tasks, we propose a novel masking strategy grounded in recent findings in the field: **(1) Task Affinity**: Denoising tasks at adjacent timesteps have a higher task affinity than those at distant timesteps (Go et al., 2023a). **(2) Task Weights**: Previous studies have shown improvements in diffusion training by assigning higher weights to denoising tasks at higher timesteps compared to lower timesteps (Hang et al., 2023; Choi et al., 2022; Go et al., 2023a). This aligns with the observation that diffusion models primarily learn perceptually rich content at higher timesteps, whereas they focus on straightforward noise removal at lower timesteps (Choi et al., 2022).

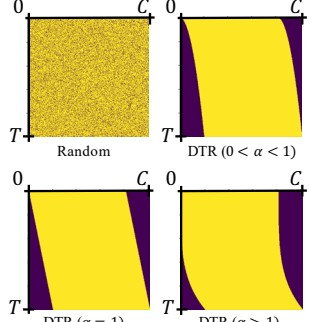

Figure 2: **Routing masks** in random routing and DTR with varying $\alpha$ ($\beta$ is fixed to 0.8). The activated and deactivated channels are colored in yellow and purple, respectively.

To integrate the concept of **(1) Task Affinity**, we employ a sliding window of size $C_\beta$ within the mask, activating channels within its boundaries. As we increase timesteps from 1 to $T$, the sliding window gradually shifts. This ensures that denoising tasks at neighboring timesteps engage similar sets of channels, while those at distant timesteps reduce channel sharing. The underlying principle here is that sharing channels between tasks having higher task affinity proves beneficial for training, as demonstrated in (Fifty et al., 2021). To incorporate **(2) Task Weights**, we use an additional parameter, $\alpha$. This modulates the shifting ratio of the sliding window across timesteps, manipulating the amount of allocation of task-dedicated channels to each timestep.

To incorporate the above two concepts, the available start index of activated channels in $\{0, \ldots C - C\beta\}$, and we quantized this start index according to timestep as $(\frac{t}{T})^\alpha$, enabling modulation of shifting ratio of the sliding window. Formally, the masks are initialized as:

$$\boldsymbol{m}_{D_t,c} = \begin{cases} 1, & \text{if } \lfloor (C - C_\beta) \cdot \left(\frac{t-1}{T}\right)^\alpha \rceil < c \leq \lfloor (C - C_\beta) \cdot \left(\frac{t}{T}\right)^\alpha \rceil + C_\beta, \\ 0, & \text{otherwise.} \end{cases} \quad (4)$$

A conceptual visualization of our masking strategy is shown in Fig. 2. When $0 < \alpha < 1$, it allocates more task-dedicated channels to smaller timesteps. At $\alpha = 1$, task-dedicated channels are evenly distributed across all timesteps. Lastly, for $\alpha > 1$, more task-dedicated channels are assigned to higher timesteps, aligning with our intent to give more weight to the higher timesteps. However, setting $\alpha$ to too large values causes the initial sliding window shift to occur at very large timesteps, in turn, leads to a situation where many tasks no longer have task-dedicated channels.

| Latent-Level Diffusion Model. Image Resolution 256×256 | | | | | | Pixel-Level Diffusion Model. Image Resolution 64×64 | | | | | |
|---|---|---|---|---|---|---|---|---|---|---|---|
| Model | FFHQ | ImageNet | | | MS-COCO | Model | FFHQ | ImageNet | | | |
| | FID↓ | FID↓ | IS↑ | Prec↑ | Rec↑ | FID↓ | | FID↓ | FID↓ | IS↑ | Prec↑ | Rec↑ |
| DiT | 10.99 | 12.59 | 134.60 | 0.73 | 0.49 | 7.70 | ADM | 3.12 | 6.34 | 72.05 | 0.85 | **0.46** |
| DiT + R-TR | 11.99 | 16.18 | 105.88 | 0.70 | **0.51** | 10.49 | ADM + R-TR | 3.33 | 7.34 | 62.62 | 0.84 | 0.44 |
| **DiT + DTR** | **7.32** | **8.90** | **156.48** | **0.77** | **0.51** | **7.04** | **ADM + DTR** | **2.65** | **4.91** | **82.44** | **0.88** | 0.46 |

Table 1: **Comparative results.** We evaluate unconditional image generation on FFHQ, class-conditional image generation on ImageNet, and text-conditional image generation on MS-COCO. We set the activation ratio $\beta$ to $0.8$ for both R-TR and DTR. Note that our DTR achieves substantial performance improvements without additional parameters or significant computational costs.

## 5 EXPERIMENTAL RESULTS

In this section, we present experimental results to validate the effectiveness of our method. To begin, we outline our experimental setups in Sec. 5.1. Then, we provide the results of a comparative evaluation in Sec. 5.2, showing that our method significantly improves FID, IS, Precision, and Recall metrics compared to the baseline. Finally, we delve into a comprehensive analysis of DTR in Sec. 5.3, dissecting its performance across multiple dimensions.

### 5.1 EXPERIMENTAL SETUP

Due to space constraints, we provide a concise overview of our experimental setups here. More extensive information regarding all our experimental settings can be found in Appendix C.

**Evaluation protocols.** To assess the effectiveness of our method, we conducted a comprehensive evaluation across three image-generation tasks: 1) Unconditional generation: we utilized FFHQ (Karras et al., 2019), which contains 70K training images of human faces. 2) Class-conditional generation: we used ImageNet (Deng et al., 2009), which contains 1,281,167 training images from 1K different classes. 3) Text-to-Image generation: we used MS-COCO (Lin et al., 2014), which contains 82,783 training images and 40,504 validation images, each annotated with 5 descriptive captions.

**Evaluation metrics.** To evaluate the quality of the generated samples, we used FID (Heusel et al., 2017), IS (Salimans et al., 2016), and Precision/Recall (Kynkäänniemi et al., 2019). Specifically, FID is used for sample quality in unconditional and text-to-image generation. Then, FID, IS, and Precision are used for sample quality measure and Recall is used for diversity measure in class-conditional generation.

**Models.** To verify the broad applicability of our method, we utilized two representative architectures: UNet-based ADM (Dhariwal & Nichol, 2021) and Transformer-based DiT (Peebles & Xie, 2022). For text-to-image generation, we used a CLIP text encoder (Radford et al., 2021) to transform textual descriptions into a sequence of embeddings for the condition of diffusion models.

### 5.2 COMPARATIVE EVALUATION

**Quantitative results.** We quantitatively validate our approach on well-established architectures, *e.g.*, DiT and ADM. The results are presented in Table 1. Firstly, we observe that naive random routing (R-TR) leads to performance degradation. This occurs because the R-TR approach lacks the capability to incorporate prior knowledge of the diffusion model (**Task Affinity**) specific to denoising tasks, as it relies on random instantiation of routing masks. In contrast, DTR incorporates the prior knowledge of denoising tasks in diffusion models, mentioned as Task Affinity and Task Weights in Sec. 4.3. Therefore, through its straightforward design, our DTR consistently demonstrates significant performance enhancements across all metrics for three datasets when compared to the model without DTR. Note that DTR achieves substantial performance improvements with no extra parameters and with negligible computational overhead for multiplications of channel masks.

**Compatibility of DTR with MTL loss weighting techniques.** In Table 2, we show that our DTR, an MTL architectural approach for diffusion models, is compatible with MTL loss weighting techniques specifically designed for diffusion models (Go et al., 2023a; Hang et al., 2023) as well as improved loss weighting method (Choi et al., 2022), both in class-conditional and unconditional

**Class-Conditional ImageNet 256×256.**

| Loss Weight Type | DiT-L/2 | | | | DiT-L/2 + DTR | | | |
|---|---|---|---|---|---|---|---|---|
| | FID↓ | IS↑ | Prec↑ | Rec↑ | FID↓ | IS↑ | Prec↑ | Rec↑ |
| Vanilla | 12.59 | 134.60 | 0.73 | 0.49 | 8.90 | 156.48 | 0.77 | **0.51** |
| Min-SNR (Hang et al., 2023) | 9.58 | 179.98 | 0.78 | 0.47 | 8.24 | 186.02 | 0.79 | 0.50 |
| ANT-UW (Go et al., 2023a) | 5.85 | 206.68 | **0.84** | 0.46 | **4.61** | **208.76** | **0.84** | 0.48 |

**Unconditional FFHQ 256×256.**

| FID-10K | Loss Weight Type | | | |
|---|---|---|---|---|
| | Vanilla | Min-SNR (Hang et al., 2023) | ANT-UW (Go et al., 2023a) | P2 (Choi et al., 2022) |
| DiT-B/2 | 12.93 | 9.73 | 9.30 | 10.08 |
| DiT-B/2 + DTR | 8.82 | 8.93 | **8.81** | 8.83 |

Table 2: **Compatibility of DTR with MTL loss weighting methods.** In a class-conditional generation, utilizing both DTR and loss weighting techniques significantly boosts performance, showing their complementarity. In unconditional generation, employing only DTR nearly matches the best performance, which underscores the effectiveness of DTR as a standalone solution.

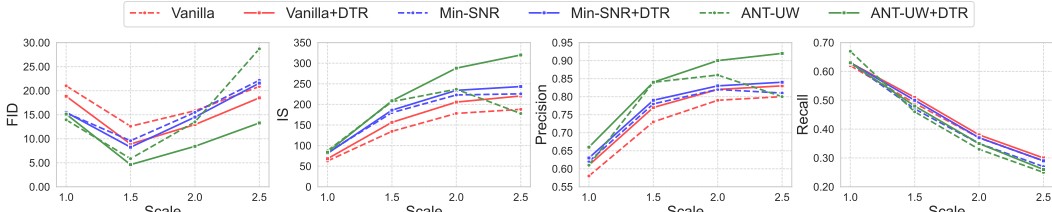

Figure 3: **Compatibility of DTR and MTL loss weighting methods w.r.t. guidance scale.** DTR robustly boosts the performance across various guidance scales for all metrics.

generation scenarios. Here, we use the DiT architecture as our baseline model. Initially, we observe that applying loss weighting techniques yields superior performance compared to using no such techniques. In a class-conditional generation, the simultaneous use of both DTR and loss weighting techniques consistently boosts performance, implying that these two approaches complement each other effectively. Furthermore, in Fig. 3, we provide insights into performance variations across different guidance scales. The results demonstrate the robustness of our DTR across guidance scales, as DTR generally enhances FID, IS, and Precision. In unconditional generation, DTR essentially takes on the role of loss weighting techniques, reducing the necessity of additional loss weighting techniques. As a result, employing only DTR leads to performance levels that are nearly equivalent to those achieved with the combination of DTR and loss weighting techniques. This underscores the effectiveness of our DTR approach as a standalone solution for unconditional generation tasks.

**Qualitative results.** Due to space limitations, we present a comprehensive set of generated examples in Appendix H for qualitative comparison. To summarize our findings, our method for training diffusion models produces images that exhibit improved realism and fidelity when compared to diffusion models without DTR.

**Further Results on More Training Iterations.** We have explored the impact of longer training on model performance. Our extensive training of DiT-L/2 + DTR with ANT-UW (Go et al., 2023a) for 2 million iterations significantly enhanced FID scores to 2.33 on ImageNet 256×256. Table 3 shows our method, despite fewer parameters and iterations, outperforms vanilla DiT-XL/2 and rivals DiT-XL after

| Method | FID↓ | Train Iters | # Params | Flops (G) |
|---|---|---|---|---|
| DiT-XL/2 | **2.27** | 7M | 675M | 118.64 |
| DiT-XL/2 | 2.55 | 2.35M | 675M | 118.64 |
| DiT-L/2 + Ours | *2.33* | **2M** | **458M** | **80.73** |

Table 3: **More Training Iterations.** Although DTR used smaller parameters, DTR shows a similar performance compared to the larger model trained over more iterations.

7 million iterations. This underscores our approach's efficiency, demonstrating dramatic improvements by integrating MTL into diffusion model architecture and optimization.

### 5.3 ANALYSIS

**Mask instantiation strategy.** Given that the routing mask of DTR is instantiated by two hyperparameters, $\alpha$ and $\beta$, we conduct ablation studies to assess the impact of varying them in Fig. 4.

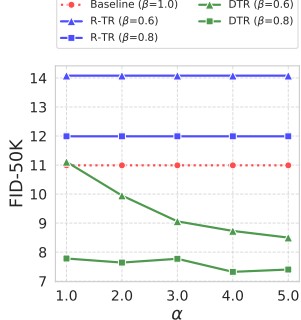

**Figure 4: $\alpha, \beta$ ablation.** We use DiT-B/2 on FFHQ 256×256.

Table 4: **Mask instantiation ablation.** We set $\beta = 0.8$ for DTR due to its stable performance, as shown in Fig. 4. In general, $\alpha = 4$ yields the best results.

| Unconditional FFHQ 256×256 | | | Class-Conditional ImageNet 256×256 | | | | |
|---|---|---|---|---|---|---|---|
| Model | $\alpha$ | FID↓ | Model | $\alpha$ | FID↓ | IS↑ | Prec↑ | Rec↑ |
| DiT-B/2 ($\beta = 1$) | - | 10.99 | DiT-L/2 ($\beta = 1$) | - | 12.59 | 134.60 | 0.73 | 0.49 |
| DiT-B/2 + DTR | 0.5 | 7.78 | DiT-L/2 + DTR | 0.5 | 11.55 | 146.06 | 0.75 | 0.49 |
|  | 1.0 | 7.78 |  | 1.0 | 10.31 | 149.23 | 0.76 | 0.49 |
|  | 2.0 | 7.64 |  | 2.0 | 9.39 | 154.06 | **0.77** | 0.49 |
|  | 3.0 | 7.77 |  | 3.0 | **8.89** | 154.97 | **0.77** | 0.49 |
|  | 4.0 | **7.32** |  | 4.0 | 8.90 | **156.48** | **0.77** | **0.51** |
|  | 5.0 | 7.40 |  | 5.0 | 9.66 | 156.47 | **0.77** | 0.50 |

| Text-Conditional MS-COCO 256×256 | | | Class-Conditional ImageNet 64×64 | | | | |
|---|---|---|---|---|---|---|---|
| Model | $\alpha$ | FID↓ | Model | $\alpha$ | FID↓ | IS↑ | Prec↑ | Rec↑ |
| DiT-B/2 ($\beta = 1$) | - | 7.70 | ADM ($\beta = 1$) | - | 6.34 | 72.05 | 0.85 | 0.46 |
| DiT-B/2 + DTR | 1.0 | 8.25 | ADM + DTR | 1.0 | 5.30 | 74.74 | 0.86 | 0.45 |
|  | 3.0 | 7.25 |  | 3.0 | 5.44 | 74.75 | 0.85 | **0.46** |
|  | 4.0 | 7.04 |  | 4.0 | **4.91** | **82.44** | **0.88** | **0.46** |
|  | 5.0 | **6.93** |  | 5.0 | 5.22 | 78.12 | 0.86 | 0.45 |

| Class-Conditional ImageNet 256×256. | | | | |
|---|---|---|---|---|
| Model | FID↓ | IS↑ | Prec↑ | Rec↑ |
| DiT-S/2 | 44.28 | 32.31 | 0.41 | 0.53 |
| DiT-S/2 + DTR | **37.43** | **38.97** | **0.47** | **0.54** |
| DiT-B/2 | 27.96 | 64.72 | 0.57 | 0.52 |
| DiT-B/2 + DTR | **16.58** | **87.94** | **0.66** | **0.53** |
| DiT-L/2 | 12.59 | 134.60 | 0.73 | 0.49 |
| DiT-L/2 + DTR | **8.90** | **156.48** | **0.77** | **0.51** |

Table 5: **Impact of DTR w.r.t. model size.** Note that DTR achieves consistent improvements across model sizes.

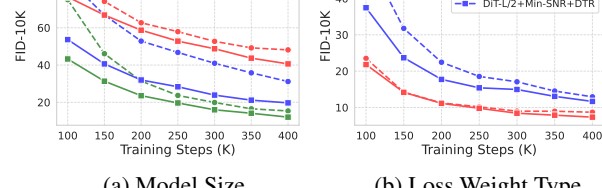

(a) Model Size      (b) Loss Weight Type

Figure 5: **Convergence comparison on ImageNet.** DTR accelerates faster FID-10K improvement.

To provide a clear context, we set a baseline (DiT-B/2 without any task routing) and include R-TR (DiT-B/2 with random routing) for comparison. We initially observe that setting $\beta$ to 0.8 rather than 0.6 leads to superior performance for both DTR and R-TR and notably, $\beta = 0.8$ exhibits robust behavior w.r.t. variations in $\alpha$. Consequently, we fix $\beta$ at 0.8.

To delve deeper into the impact of $\alpha$ on performance, we report the results by changing $\alpha$ on each dataset in Table 4. Increasing $\alpha$ corresponds to allocating more channels to tasks at higher timesteps. Our results indicate that $\alpha = 4$ yields the best performance across almost all datasets and evaluation metrics. Note that increasing $\alpha$ leads to significant performance improvements up to a certain threshold ($\alpha = 4$), beyond which performance begins to degrade. This suggests that allocating a moderately larger capacity to tasks at higher timesteps is beneficial for overall performance. This suggests that our design principle for DTR, **Task Weight**s – allocating a moderately larger capacity to tasks at higher timesteps, is beneficial for overall performance. From the results, we opt to fix $\beta$ at 0.8 and $\alpha$ at 4 for further evaluations.

**Impact of DTR with respect to model size.** In Table 5, we present the results of a controlled scaling study of DiT on the ImageNet dataset, focusing on how DTR affects the performance according to the DiT model sizes (S, B, L). Initially, we observe considerable performance improvements as we scale up the model. Importantly, applying DTR further enhanced performance across all model sizes, with larger models benefiting more. It is hypothesized because as the model size increases, the total number of channels increases, thus more task-dedicated channels can be allocated.

**Convergence speed.** In Fig. 5, we present a comparative analysis of convergence speed, focusing on the impact of training with and without DTR. First, we investigate training convergence by integrating DTR into DiT models of varying sizes (S, B, L). As shown in the results, the addition of DTR leads to a significant speed boost regardless of the model size. In particular, training DiT-B/2 without DTR takes roughly 400K training iterations to reach an FID score of 31, whereas, with DTR, it achieves the same result in only 200K iterations, effectively doubling the speed. Additionally, we explore the synergy between DTR and MTL loss weighting methods (ANT-UW (Go et al., 2023a) and Min-SNR (Hang et al., 2023)), which can boost the convergence of DiT, in the context of DiT-L/2. While using MTL loss weighting methods alone provides a certain degree of convergence acceleration, integrating DTR can further enhance convergence speed. Moreover, DTR mitigates the

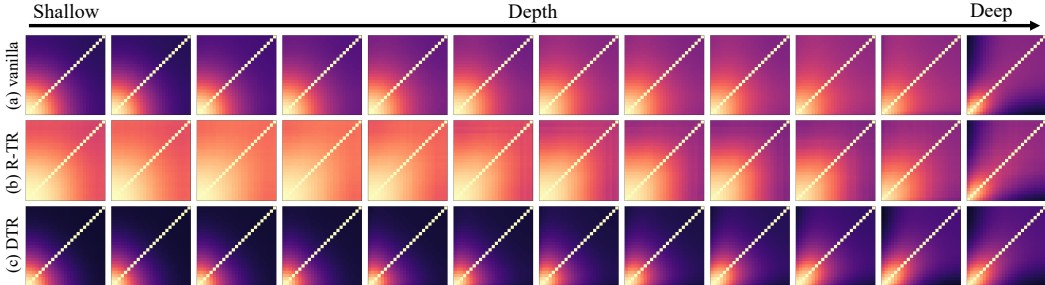

Figure 6: **Comparison of CKA representation similarity on FFHQ dataset**. We show how inter-task representation similarity changes when applying task routing in 12 DiT blocks. The horizontal and vertical axes in each plot represent timesteps (from $t = 1$ to $t = T$). We assess three configurations: (a) DiT alone *vs.* (b) DiT with R-TR *vs.* (c) DiT with DTR. R-TR generally increases CKA similarity, whereas DTR decreases it. Brighter/darker color represents higher/lower similarity.

saturation issue that emerges at 300K in DiT-L/2 + ANT-UW, leading to a more stable convergence and accelerated learning. Through this, we confirm that explicitly handling negative transfer with DTR can significantly improve training dynamics.

**Representation analysis via CKA.** In Fig. 6, we employ Centered Kernel Alignment (CKA) (Kornblith et al., 2019) to visualize the similarity between intra-model representations. Specifically, we examine how similar or different the model representations are at different timesteps within each DiT block to gain insights into the model's behavior. CKA quantifies this similarity, where a higher score indicates that the model behaves similarly across different timesteps, while a lower score implies that the model's behavior varies significantly across different timesteps.

We make noteworthy observations by comparing three scenarios: DiT model without task routing *vs.* DiT with random routing *vs.* DiT with DTR: (1) Upon introducing random routing to the DiT model, we notice an overall increase in CKA compared to the baseline. This implies that with random routing, the model's representations remain more similar across various timesteps. (2) With DTR, we observe a distinct pattern in the CKA scores, where there are high scores at lower timesteps and low scores at higher timesteps. We can interpret that at higher timesteps, the model primarily focuses on learning discriminative features that are relevant to specific timesteps, whereas at lower timesteps, the model tends to exhibit similar behavior across different timesteps. This aligns with our design principle, **Task Weights**. (3) In the later blocks with DTR, there is a notable highlight on diagonal elements. This suggests that the model takes into account **Task Affinity**, reflecting the model's ability to make its behavior more similar for adjacent timesteps.

**Comparison to multi-experts strategy.** We compare DiT-L/2 equipped with DTR against a multi-experts model (DiT-B/2 × 4), with each expert specializing in certain timesteps, *e.g.*, $[0, T/4], \dots, [3T/4, T]$. Here, we show that DTR outperforms the multi-experts denoiser method (Balaji et al., 2022; Lee et al., 2023). For detailed results, please refer to Appendix D.

## 6 DISCUSSION

In this work, we have proposed DTR, a simple add-on strategy for diffusion models that establishes task-specific pathways within a single model while embedding prior knowledge of denoising tasks into the model through *explicit* architectural modifications. Our experimental findings clearly indicate that DTR represents a significant leap forward compared to current diffusion model architectures, which rely solely on *implicit* signals. Importantly, this improvement is achieved only with a negligible computational cost and without introducing additional parameters. Our work conveys two important messages: (1) We found that relying solely on *implicit* signals for enhancing task adaptability of diffusion models, *e.g.*, conditioning on timesteps or noise levels, proves insufficient to mitigate negative transfer between denoising tasks. (2) By *explicitly* addressing the issue of negative transfer and incorporating prior knowledge of denoising tasks into diffusion model architectures, our work shows promise in enhancing their performance across various generative tasks. To the best of our knowledge, our study is the first to advance diffusion model architecture from an MTL perspective. We hope our work will inspire further investigations in this direction.

## 7 ETHICS STATEMENT

Generative models, such as diffusion models, have the potential to exert profound societal influence, with particular implications for deep fake applications and the handling of biased data. A critical focus is on the potential amplification of misinformation and the erosion of trust in visual media. In addition, when generative models are trained on datasets with biased or deliberately manipulated content, there is the unintended consequence of inadvertently reinforcing and exacerbating social biases, thereby facilitating the spread of deceptive information and the manipulation of public perception. We will encourage the research community to discuss ideas to prevent these unintended consequences.

## 8 REPRODUCIBILITY STATEMENT

We present details on implementation and experimental setups in our main manuscripts and Appendix. To further future works from our work, we release our experimental codes and checkpoints at https://github.com/byeongjun-park/DTR.

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

# APPENDIX

## CONTENTS

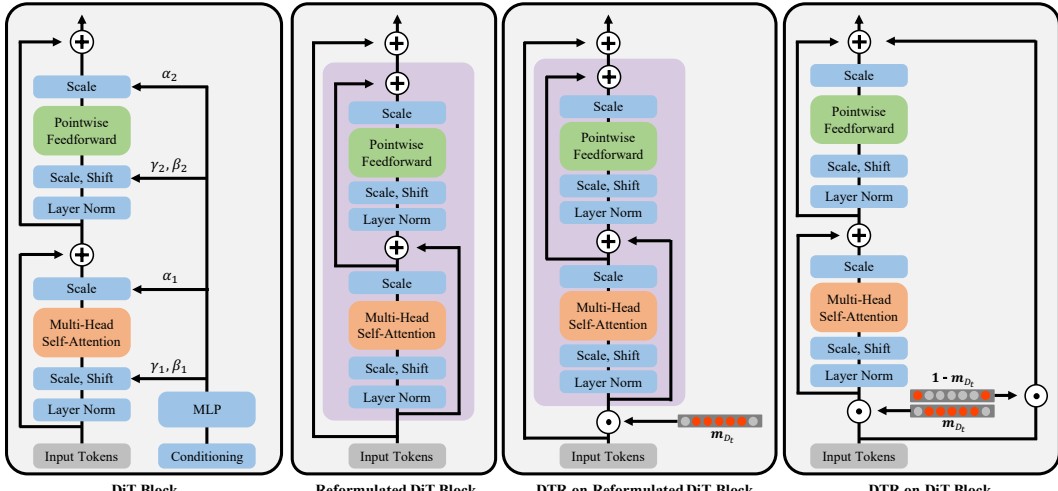

Figure 7: **DiT block with DTR.** $\odot$ represents an element-wise multiplication. We only show the conditioning block in the DiT block (leftmost), as all blocks use the same conditioning block.

## A IMPLEMENTATION DETAILS ON DENOISING TASK ROUTING

In this section, we present the implementation details on Denoisng Task Routing (DTR). Firstly, in Sec. A.1, we describe the details of implementation for incorporating DTR in ADM (Dhariwal & Nichol, 2021) and DiT (Peebles & Xie, 2022) architectures. To provide further details, we illustrate pseudocode for the task routing mechanism and the routing mask instantiation in Sec. A.2.

### A.1 IMPLEMENTATION DETAILS ON ADM AND DIT

**ADM (Dhariwal & Nichol, 2021)** We apply DTR on two types of residual blocks, an Attention (Vaswani et al., 2017) block, and a ResNet (He et al., 2016) block. Due to the potential effect of the channel masking on local running mean and variance, DTR is positioned right after the group normalization layer (Wu & He, 2018) in both types of residual blocks. Then, we can easily apply DTR since ADM uses the same configured residual block in Eq. (3).

**DiT (Peebles & Xie, 2022)** DiT introduces a zero-initialized adaptive layer normalization (adaLN-Zero) in transformer blocks, where the normalization parameters are regressed from the timestep embeddings and conditions. While DiT also utilizes residual connections within the transformer block, it does not directly adhere to the formulation outlined in Eq. (3). Consequently, we reformulate the DiT block, introducing the necessary modifications to integrate DTR.

The original $l$-th DiT block $\text{DiTBlock}^l$ outputs $\boldsymbol{z}^{l+1}$ given the input $\boldsymbol{z}^l$ as:

$$\boldsymbol{z}^{l+1} = \text{DiTBlock}^l(\boldsymbol{z}^l) = (\boldsymbol{z}^l + \text{Attn}^l(\boldsymbol{z}^l)) + \text{MLP}^l(\boldsymbol{z}^l + \text{Attn}^l(\boldsymbol{z}^l)), \qquad (5)$$

where $\text{Attn}^l$ and $\text{MLP}^l$ represent a multi-head self-attention and a pointwise feedforward layer in $l$-th DiT block, both including adaLN-Zero layers. We note that this can be regarded as a residual block by defining the block as:

$$\text{Block}^l(\boldsymbol{z}^l) = \text{Attn}^l(\boldsymbol{z}^l) + \text{MLP}^l(\boldsymbol{z}^l + \text{Attn}^l(\boldsymbol{z}^l)). \qquad (6)$$

The left two blocks in Fig. 7 show an overview of how we reformulate the DiT block. We then apply DTR on the reformulated DiT block by using Eq. (3) and Eq. (6), which is expanded as:

$$\boldsymbol{z}^{l+1} = \boldsymbol{z}^l + \text{Attn}^l(\boldsymbol{m}_{D_t} \odot \boldsymbol{z}^l) + \text{MLP}^l((\boldsymbol{m}_{D_t} \odot \boldsymbol{z}^l) + \text{Attn}^l(\boldsymbol{m}_{D_t} \odot \boldsymbol{z}^l)). \qquad (7)$$

Here, using the existing DiT block in Eq. (5), it can be expressed as:

$$\boldsymbol{z}^{l+1} = (1 - \boldsymbol{m}_{D_t}) \odot \boldsymbol{z}^l + \text{DiTBlock}^l(\boldsymbol{m}_{D_t} \odot \boldsymbol{z}^l). \qquad (8)$$

Interestingly, this can be implemented simply by applying DTR to the original DiT block and then incorporating a skip connection from the output with a complementary routing mask at the end.

---

**Pseudo Code 1** [NumPy-like] Random Masking (Left) *vs.* DTR Masking (Right)

```
1   # T: number of tasks
2   # C: number of channels
3   # beta: activation ratio
4
5   def random_masking(T, C, beta):
6       # initialize mask with zeros
7       mask = np.zeros(T, C)
8       # number of activated channels
9       num_activ = int(beta * C)
10      # fill the mask with ones
11      mask[:, : num_activ] = 1
12      # randomly shuffle the columns of the mask
13      return mask[:, np.random.permutation(C)]
```

```
1   # T: number of tasks
2   # C: number of channels
3   # alpha: channel shifting parameter
4   # beta: activation ratio
5
6   def dtr_masking(T, C, alpha, beta):
7       # initialize mask with zeros
8       mask = np.zeros(T, C)
9       # number of activated channels
10      num_activ = int(beta * C)
11      # number of deactivated channels
12      num_deact = C - num_activ
13      # create linearly spaced points
14      x = np.linspace(0, 1, T)
15      # apply a scaling factor to linear points
16      x = x ** alpha
17      # calculate the channel offset for every timesteps
18      offset = (num_deact * x).round()
19      # fill the mask with ones
20      for t in T:
21          start = offset[t]
22          end = offset[t] + num_activ
23          mask[t, start:end] = 1
24      return mask
```

---

**Pseudo Code 2** [Simplified] ADM block (Left) *vs.* ADM block + DTR (Right)

```
1   # z: input representation
2
3   def forward(z):
4       # apply normalization, SiLU
5       h = SiLU(norm(z))
6       # up-interpolation + conv
7       h = conv(upsample(h))
8       # apply conv
9       h = conv(h)
10      # apply normalization, SiLU, conv
11      h = conv(SiLU(norm(h)))
12      # add original representation
13      return conv(upsample(z)) + h
```

```
1   # z: input representation
2   # mask: routing mask
3
4   def forward(z, mask):
5       # apply normalization, SiLU
6       h = SiLU(norm(z))
7       # apply the routing mask
8       m_z = mask * h
9       # up-interpolation + conv
10      m_z = conv(upsample(m_z))
11      # apply conv
12      m_z = conv(m_z)
13      # apply normalization, SiLU, conv
14      m_z = conv(SiLU(norm(m_z)))
15      # add original representation
16      return conv(upsample(z)) + m_z
```

---

**Pseudo Code 3** [Simplified] DiT block (Left) *vs.* DiT block + DTR (Right)

```
1   # z: input representation
2
3   def forward(z):
4       # apply norm, attention, skip connection
5       z = z + attention(norm1(z))
6       # apply norm, mlp, skip connection
7       z = z + mlp(norm2(z))
8       return z
```

```
1   # z: input representation
2   # mask: routing mask
3
4   def forward(z, mask):
5       # apply the routing mask
6       m_z = mask * z
7       # apply norm, attention, skip connection
8       m_z = m_z + attention(norm1(m_z))
9       # apply norm, mlp, skip connection
10      m_z = m_z + mlp(norm2(m_z))
11      # add original representation with complement mask
12      return (1-mask) * z + m_z
```

---

## A.2  PSEUDOCODE

Our DTR is easy to implement yet highly effective. Adding just a few lines of code can lead to a significant performance boost. This can be observed in the pseudocode examples. These code snippets illustrate the concept of random masking and the implementation of masking using the DTR (see Pseudo Code 1). To provide further clarity, we also offer pseudocode for a simplified version of

the ADM ResBlock and the DiT block, both extended with DTR functionality (see Pseudo Code 2 and Pseudo Code 3, respectively).

## B    THE AVERAGE PORTION OF SHARED CHANNELS IN RANDOM MASKING STRATEGY

To verify that the random masking does not take into account the relationships between tasks, we derive the expected value $\mathbb{E}(X)$ of the shared channel, where $X$ is a random variable representing the number of shared channels for two tasks $D_{t_i}$ and $D_{t_j}$. For ease of understanding, we abbreviate two tasks $D_{t_i}$ and $D_{t_j}$ as $i$ and $j$. Intuitively, when $i = j$, the all channels are shared, yielding $\mathbb{E}(X) = C_\beta$. In the case of $i \neq j$, without loss of generality, we first sample the channel indices set $\mathrm{R}(i) = \{i_1, ..., i_{C_\beta}\} \subset \{1, ..., C\}$ for task $i$. When selecting $\mathrm{R}(j)$, the probability $\mathrm{P}(k)$ of selecting $k$ shared channels from $\mathrm{R}(i)$ and selecting the rest from others is $\binom{C_\beta}{k}\binom{C-C_\beta}{C_\beta-k}/\binom{C}{C_\beta}$. Finally, the expectation value of $X$ is derived as follows:

$$\mathbb{E}(X) = \begin{cases} C_\beta, & \text{if } i = j, \\ \Sigma_{k=1}^{C_\beta} k\mathrm{P}(k), \text{ where } \mathrm{P}(k) = \binom{C_\beta}{k}\binom{C-C_\beta}{C_\beta-k}/\binom{C}{C_\beta} & \text{otherwise.} \end{cases} \quad (9)$$

Note that the expectation value of two distinct tasks remains consistent, which indicates that the randomly initialized routing mask falls short of representing the inter-task relationship as it assumes that all denoising tasks are equally related. As extensively studied in previous studies (Go et al., 2023a), **Task Affinity** is one of the prior knowledge in the field of diffusion models. This underlines why the use of random routing methods leads to a noticeable degradation in performance.

## C    DETAILED EXPERIMENTAL SETUP IN SECTION 5

**Training details.**    We employed the AdamW optimizer (Loshchilov & Hutter, 2019) with a fixed learning rate of 1e-4. No weight decay was applied during training. A batch size of 256 was used and a horizontal flip was applied to the training data. We utilized classifier-free guidance (Ho & Salimans, 2022) with a guidance scale set to 1.5 in conditional generation settings such as text-to-image generation and class-conditional image generation. For the FFHQ dataset (Karras et al., 2019), we trained for 100k iterations and evaluated model performance on 50K samples. On the ImageNet dataset (Deng et al., 2009), we trained for 400K iterations and evaluated models using 50K samples. In experiments on MS-COCO dataset (Lin et al., 2014), we trained for 400K iterations and evaluated model performance on 50K samples.

The diffusion timestep $T$ was set to 1,000 for training and DDPM 250-step (Ho et al., 2020) for sample generation. We used a cosine scheduling strategy (Nichol & Dhariwal, 2021) and applied an exponential moving average (EMA) to the model's parameters with a decay of 0.9999 to enhance stability. All the models were trained on 8 NVIDIA A100 GPUs. We implemented the task routing on the official code of DiT[1] and ADM[2].

For implementing loss weighting methods such as Min-SNR (Hang et al., 2023), ANT-UW (Go et al., 2023a), and P2 (Choi et al., 2022), we used the officially released code of Min-SNR[3] and P2[4] for implementing their weighting method. However, since ANT does not release their code, we re-implemented ANT-UW with timestep-based clustering when the number of clusters is 8.

**Evaluation metrics.**    We evaluated diffusion models using FID (Heusel et al., 2017), IS (Salimans et al., 2016), and Precision/Recall (Kynkäänniemi et al., 2019). Lower FID indicates a closer distribution match between generated and real data, suggesting higher quality and diversity of generated samples. Higher IS implies that the generated data are of higher quality and diversity. Precision measures whether generated images fall within the estimated manifold of real images, while Recall measures the reverse. Higher Precision and Recall reflect better alignment between the generated

---

[1] https://github.com/facebookresearch/DiT
[2] https://github.com/openai/guided-diffusion
[3] https://github.com/TiankaiHang/Min-SNR-Diffusion-Training
[4] https://github.com/jychoi118/P2-weighting

**Class-Conditional ImageNet 256×256.**

| Method | # of Parameters | FID↓ | IS↑ | Prec↑ | Rec↑ |
|---|---|---|---|---|---|
| Vanilla (DiT-L/2) | 458M | 12.59 | 134.60 | 0.73 | 0.49 |
| Multi-experts (DiT-B/2 × 4) | 521M | 9.48 | 149.79 | 0.75 | **0.51** |
| DTR (DiT-L/2) | 458M | **8.90** | **156.48** | **0.77** | **0.51** |

Table 6: **Comparison between DTR and multi-expert strategy.** Although DTR used smaller parameters, DTR outperforms the multi-expert strategy in terms of FID, IS, and Precision.

and real data distribution. We followed the evaluation protocol of ADM and used codebase[5]. Unless otherwise stated, FID is calculated with 50K generated samples.

**Additional architectural details for DiT.** Since the official DiT code only offers the implementation for class-conditional generation, we have extended it to include implementations for unconditional and text-conditional generation.

For the unconditional generation, we set the number of classes to one following the recommendation of the authors of DiT [6]. For the text-conditional generation, we utilize text tokens from CLIP (Radford et al., 2021) text encoder to condition the diffusion model. Figure 8 briefly shows the implemented DiT block for the text-conditional generation. Given that we utilize conditions as a sequence of text tokens, as opposed to a single token in the class-conditional generation, the parameters of adaLN-Zero are solely regressed from the timestep embeddings. To condition text tokens, we incorporate a multi-head cross-attention layer within the DiT block. This layer follows the same structural design as the multi-head self-attention, with text tokens serving as keys and values in the cross-attention layer (Vaswani et al., 2017). Note that optimizing unconditional and text-conditional DiT blocks is beyond the scope of our current focus, leaving opportunities for further improvement.

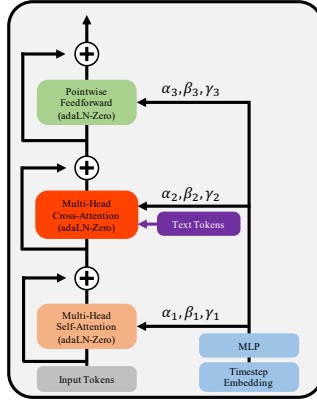

Figure 8: **DiT block for text-to-image generation.**

For all DiT experiments, we employ a VAE encoder/decoder from Stable Diffusion[7] to obtain the latent feature of input images. This VAE maps $256 \times 256 \times 3$ images into a compact latent representation with $32 \times 32 \times 4$ dimension.

## D  COMPARISON TO MULTI-EXPERTS STRATEGY

Although our works focus on effectively building a single neural network for diffusion models, comparing our DTR to works using multiple neural networks (Balaji et al., 2022; Go et al., 2023b; Lee et al., 2023) for diffusion models can further support the effectiveness of our method.

Regarding this, we compare multi-experts and DTR, when using a similar number of parameters. For constructing multi-experts, we used four DiT-B/2 models, and each model trained on a specific area of the four parts of timesteps $\{1, \ldots, T\}$. We trained each model with 200K iterations with a learning rate of 1e-4 and batch size of 256. Each DiT-B/2 model has $\approx$130.3M parameters, total used parameters for multi-experts are 521.2M parameters. For comparison above multi-experts with DTR, we used the DiT-L/2 model which has 458M parameters and was used in experiments in Sec. 5.

Table 6 shows the results of the comparison between DTR and multi-experts strategy on the ImageNet 256x256 dataset. As shown in the results, both the multi-experts strategy and DTR outperform vanilla training. Notably, our DTR outperforms the multi-experts strategy in terms of FID, IS, and Precision. This result implies that explicitly handling negative transfer in a single model with DTR can outperform parameter-separated models for covering denoising tasks.

---

[5] https://github.com/openai/guided-diffusion/tree/main/evaluations
[6] https://github.com/facebookresearch/DiT/issues/18#issuecomment
[7] https://huggingface.co/stabilityai/sd-vae-ft-ema-original

# E    COMPARISON OF COMPUTATIONAL COMPLEXITY.

Despite not requiring additional parameters, DTR incurs minimal computational cost for channel masking. To clarify this cost, we report floating point operations (FLOPs) and average training iterations executed per second across different model sizes (S, B, L) of DiT in Tab. 7. The results show a negligible increase in GFLOPs and a corresponding decrease in average training speed. This supports the computational efficiency of our DTR, demonstrating that it requires only marginal computation from its adoption to existing models.

| Method | GFLOPs | Iters/sec |
|---|---|---|
| DiT-S/2 | 6.06 | 8.19 |
| DiT-S/2 + DTR | 6.06 | 8.14 |
| DiT-B/2 | 23.01 | 6.90 |
| DiT-B/2 + DTR | 23.02 | 6.87 |
| DiT-L/2 | 80.71 | 3.72 |
| DiT-L/2 + DTR | 80.73 | 3.71 |

Table 7: **Computation comparison.**

# F    POTENTIAL ALTERNATIVES OF SLIDING WINDOW

| DiT-B/2 | Masking Strategy Type | | | | |
|---|---|---|---|---|---|
| | Vanilla | MaxRoaming (Pascal et al., 2021) | ERCDT | CDTR | DTR |
| FID↓ | 10.99 | 39.90 | 10.13 | 9.61 | **7.32** |

Table 8: **Masking Strategy Alternatives.** DTR outperforms several masking strategy alternatives, which fall short of adequately incorporating task weights or task affinity.

Here, to further support the effectiveness of DTR, we compare more extensive baselines. First, we choose MaxRoaming (Pascal et al., 2021) which utilizes the optimization strategy on randomly initialized channel masks. Second, we employ timestep-based clustering Go et al. (2023a) on DTR for validating the effects of our masking strategy on fine-grained denoising tasks compared to clustering these tasks. We used $k = 8$ of cluster size for timestep-based clustering, and initialized masks regarding each cluster as one task. We denote this as CDTR. Third, we explicitly route with clustered denoising tasks (ERCDT), where half of the channels are shared across all denoising tasks, while the remaining channels are segmented and activated for specific tasks. Comparing ERCDT with DTR can also validate the effects of whether clustering is applied or not. We trained DiT-B/2 on the FFHQ dataset using each strategy, and we present the results in Table 8.

The results show that our DTR significantly outperforms all masking strategy alternatives. The alternatives show suboptimal performance due to their failure to incorporate the diffusion prior to task weight and task affinity. For MaxRoaming, we suggest that this phenomenon is due to the detrimental effects of introducing randomness into the prior. As illustrated by ANT (Go et al., 2023a), the randomness causes negative impacts on performance, and Randomness in MaxRoaming also causes this performance degradation. For the other two alternatives, CDTR and ERCDT, the primary reason for this discrepancy lies in the inability of alternative methods to adequately capture and reflect nuanced, proximal relationships among denoising tasks inherent in the clustering approach. For example, the denoising task at $t = 1$ is considered nearly equivalent to tasks at $t = 5$ or $t = 100$ within the context of $k = 8$ clusters, failing to recognize the higher affinity between tasks at closer time intervals. Furthermore, while tasks at $t = 124$ and $t = 125$ belong to the same cluster, timesteps $t = 125$ and $t = 126$ fall into different clusters, not effectively reflecting the one-timestep difference between them. This limitation impedes the performance of ERCDT and CDTR compared to DTR, reinforcing the effectiveness of our proposed method. It is noteworthy that both CDTR and ERCDT outperform vanilla training, indicating that task routing, even with discrete representations through task clustering, enhances performance by incorporating relationships among denoising tasks.

# G    LIMITATIONS AND FUTURE WORKS

In this work, we have proposed fixed task-specific masks that incorporate the prior knowledge of denoising tasks in diffusion models. Although we showed that these fixed masks can achieve dramatic performance improvements, task-specific masks are not changed and optimized through training procedures. Despite the immutability of masks having advantages in training speed and computation,

further optimization can be more beneficial. By utilizing well-known methods such as reinforcement learning and evolutionary algorithms, the masks can be more optimized than our DTR masks and these can be future work from our work. Additionally, starting from our work, another future study could be to architecturally consider resource partitioning among multiple denoising tasks.

# H  QUALITATIVE RESULTS

## H.1  QUALITATIVE RESULTS FOR COMPARATIVE EVALUATION

**Qualitative Comparison on FFHQ Dataset**    Figure 9 shows the qualitative comparison of results on unconditional facial image generation between baseline, R-TR, and DTR. Our proposed method has better performance in generating realistic images.

**Qualitative Comparison on ImageNet Dataset**    For the comparison of conditional image generation, we show the generated results from baseline, R-TR, and DTR. As illustrated in Fig. 10, our method outperforms others.

**Qualitative Comparison on MS-COCO Dataset**    To further verify the effectiveness of the proposed method, we compare the qualitative results of the Text-to-Image generation task between baselines, R-TR, and DTR in Fig. 11.

## H.2  QUALITATIVE RESULTS FROM DiT-L/2 WITH DTR AND ANT-UW

Figures 12, 13, 14, 15 illustrates the generated images by DiT-L with DTR and ANT-UW trained on 400K iterations. As shown in the results, highly realistic images are generated by our DTR and ANT-UW despite the model being only trained on 400K iterations with a batch size of 256.

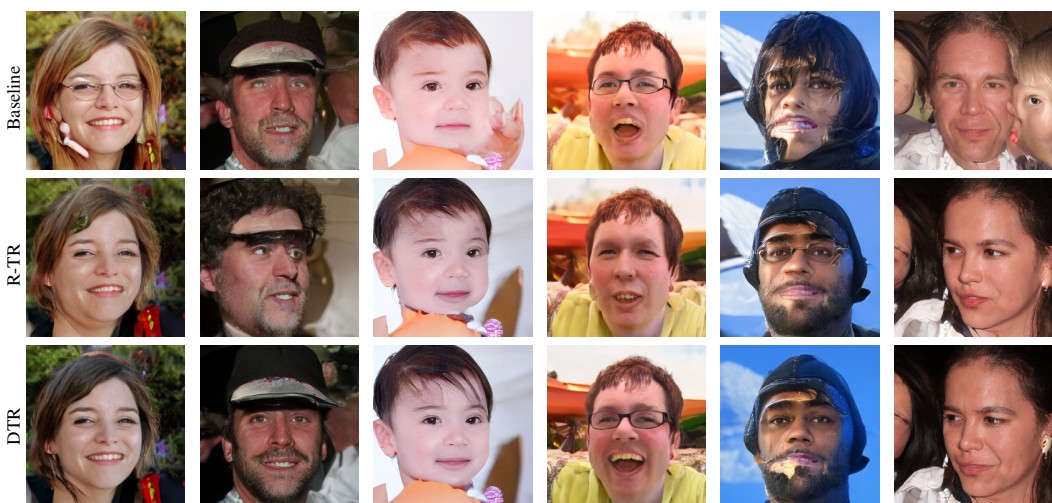

Figure 9: **Qualitative comparison between baseline, random routing (R-TR), and denoising task routing (DTR) on FFHQ dataset.**

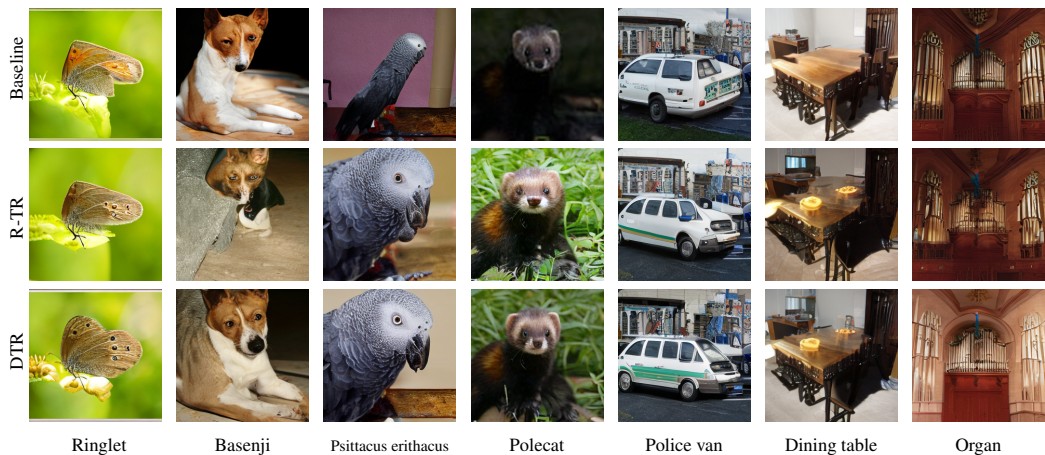

Figure 10: **Qualitative comparison between baseline, random routing (R-TR), and denoising task routing (DTR) on ImageNet dataset.**

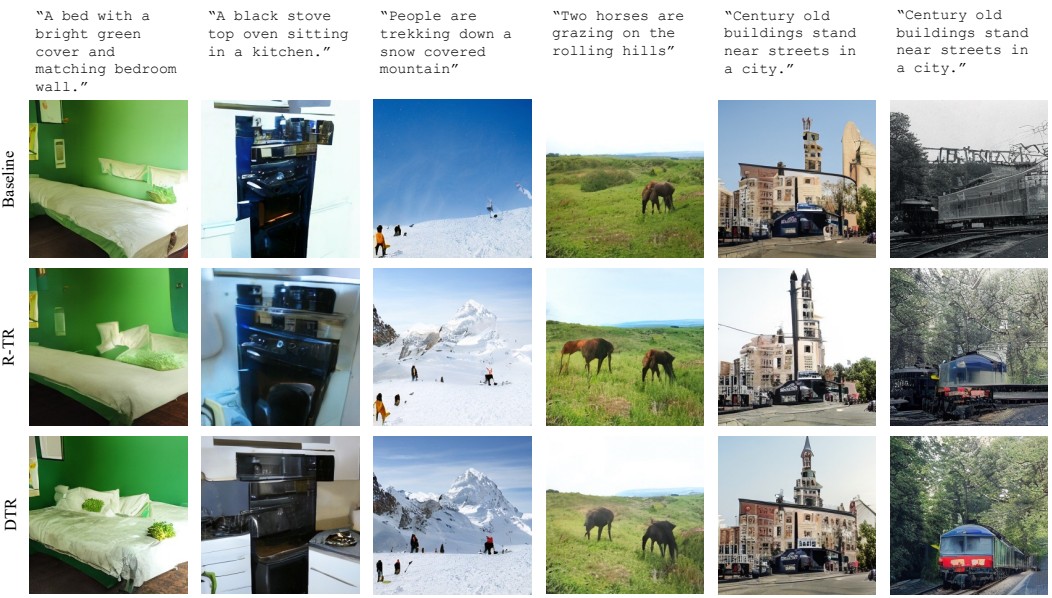

Figure 11: **Qualitative comparison between baseline, random routing (R-TR), and denoising task routing (DTR) on MS-COCO dataset.**

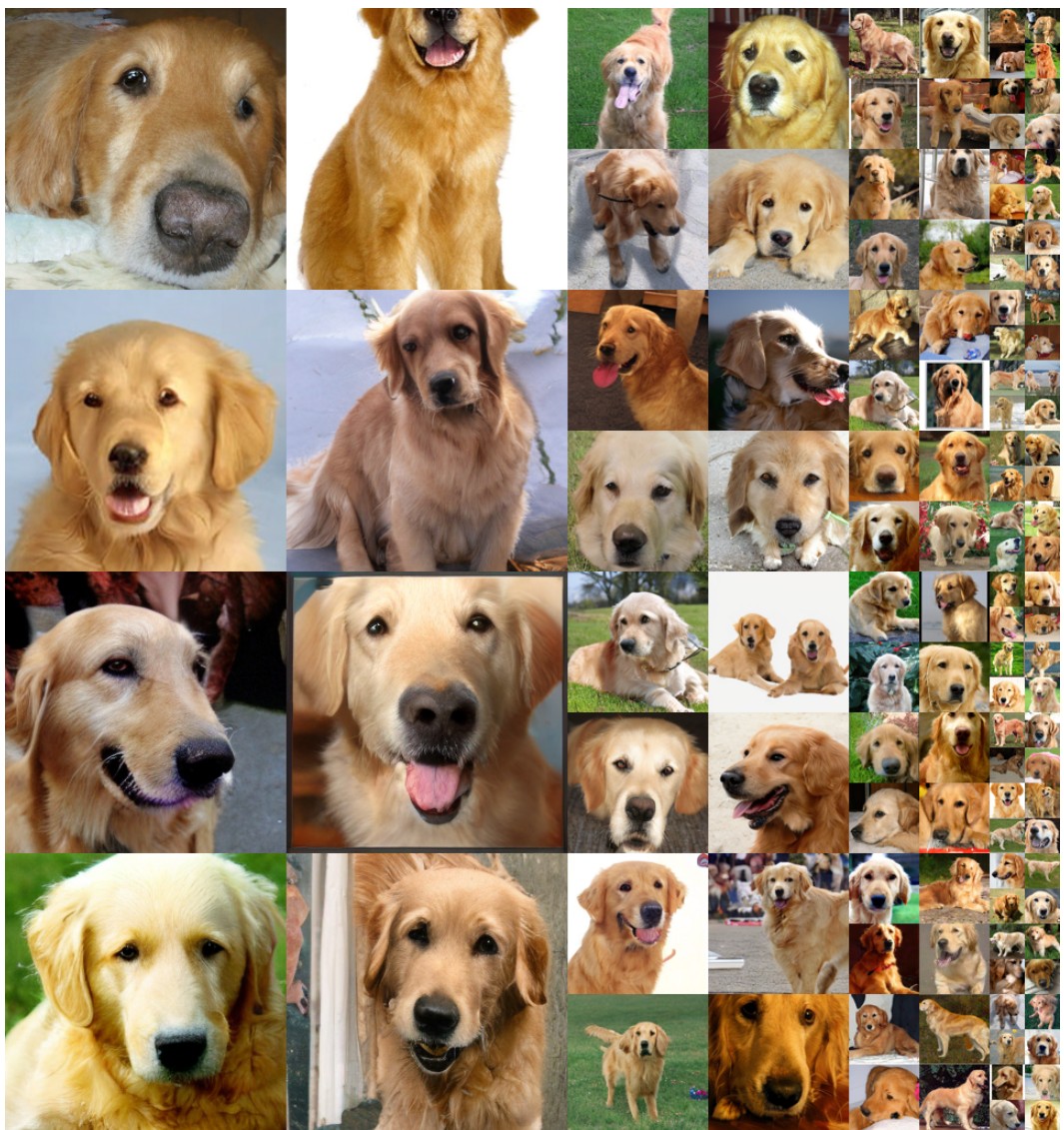

Figure 12: **Uncurated 256×256 DiT-L/2 samples.**
Classifier-free guidanzce scale = 2.0.
Class label = "golden retriever" (207)

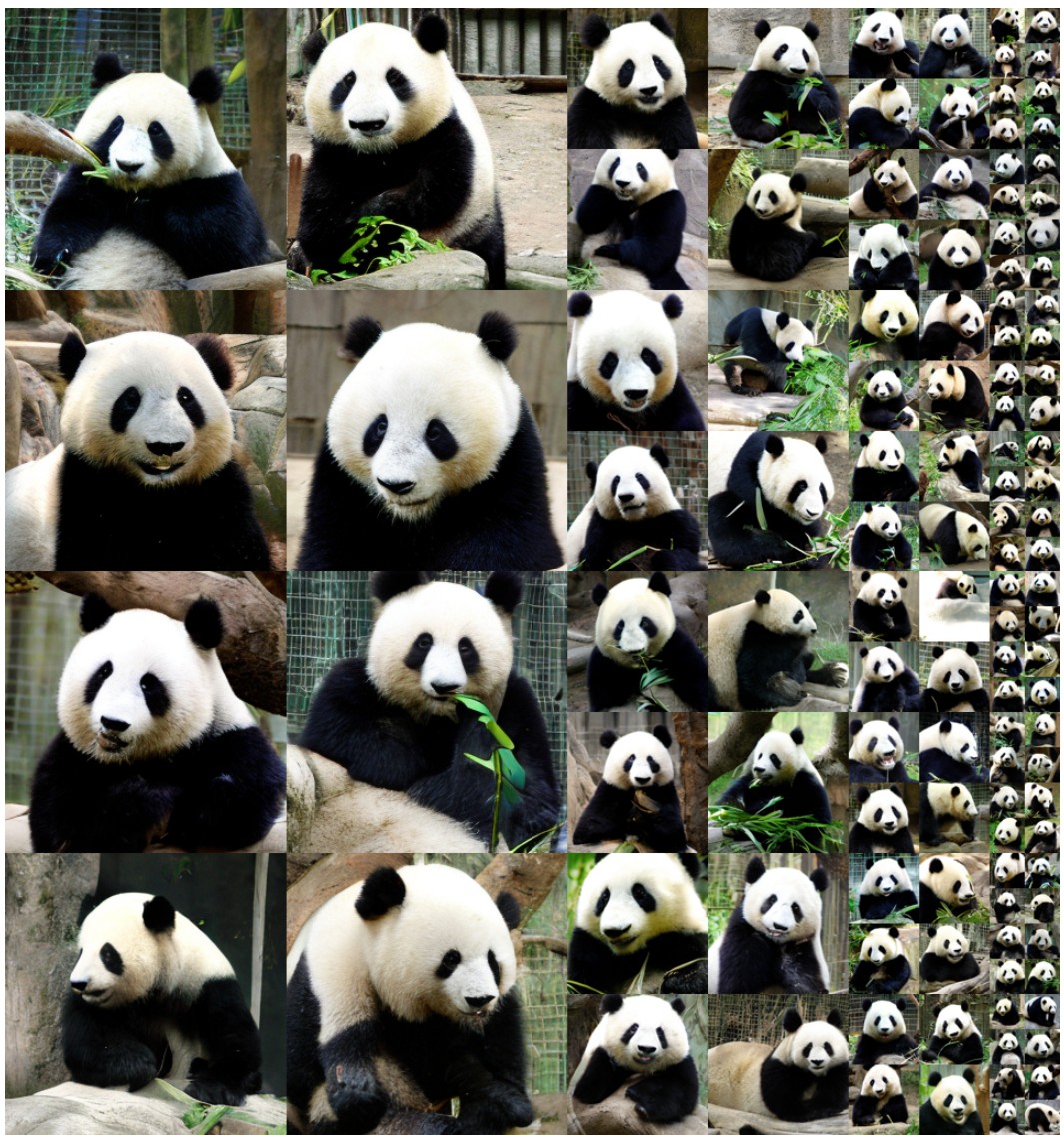

Figure 13: **Uncurated 256×256 DiT-L/2 samples.**
Classifier-free guidance scale = 2.0.
Class label = "panda" (388)

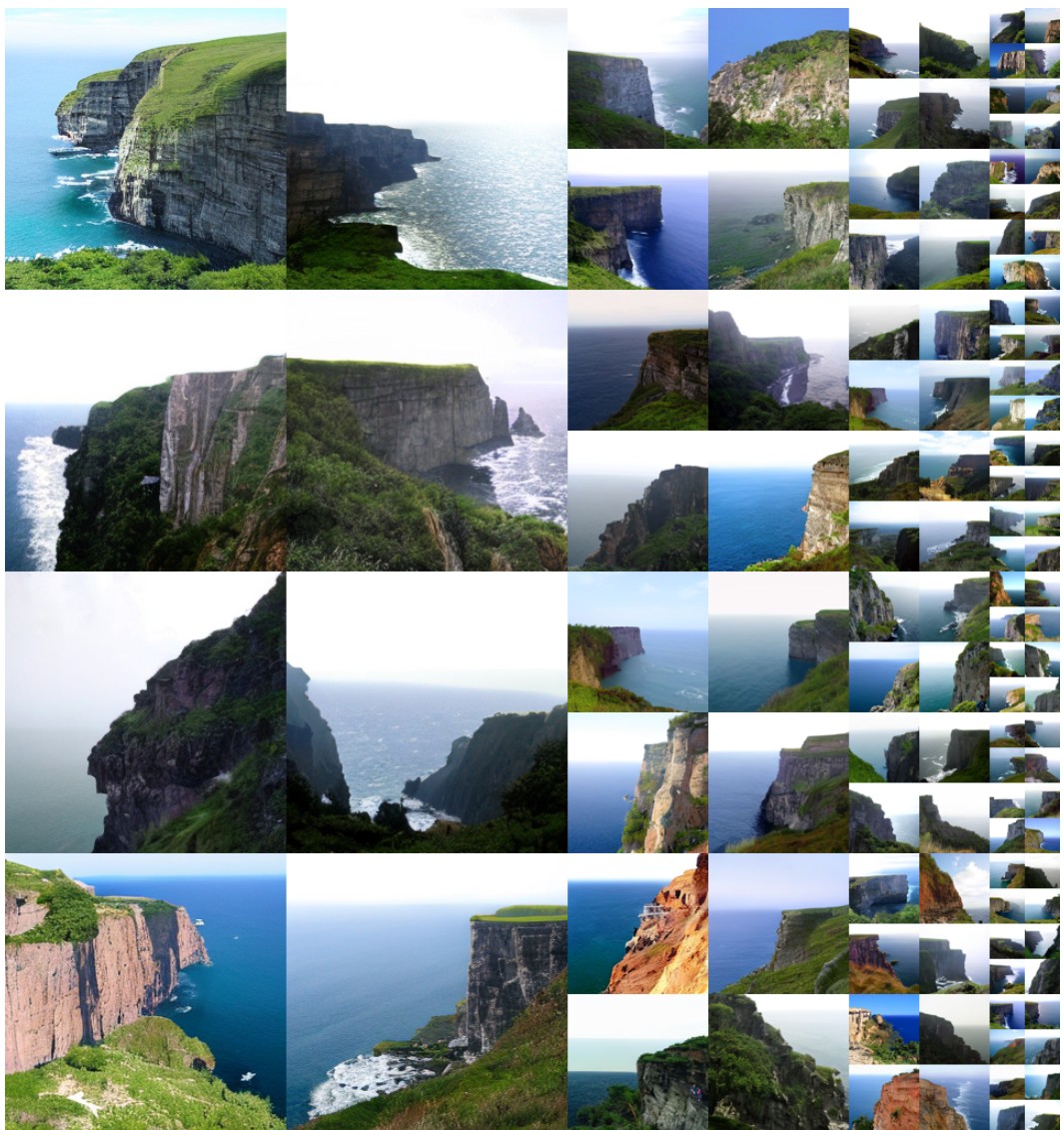

Figure 14: **Uncurated 256×256 DiT-L/2 samples.**
Classifier-free guidance scale = 4.0.
Class label = "cliff drop-off" (972)

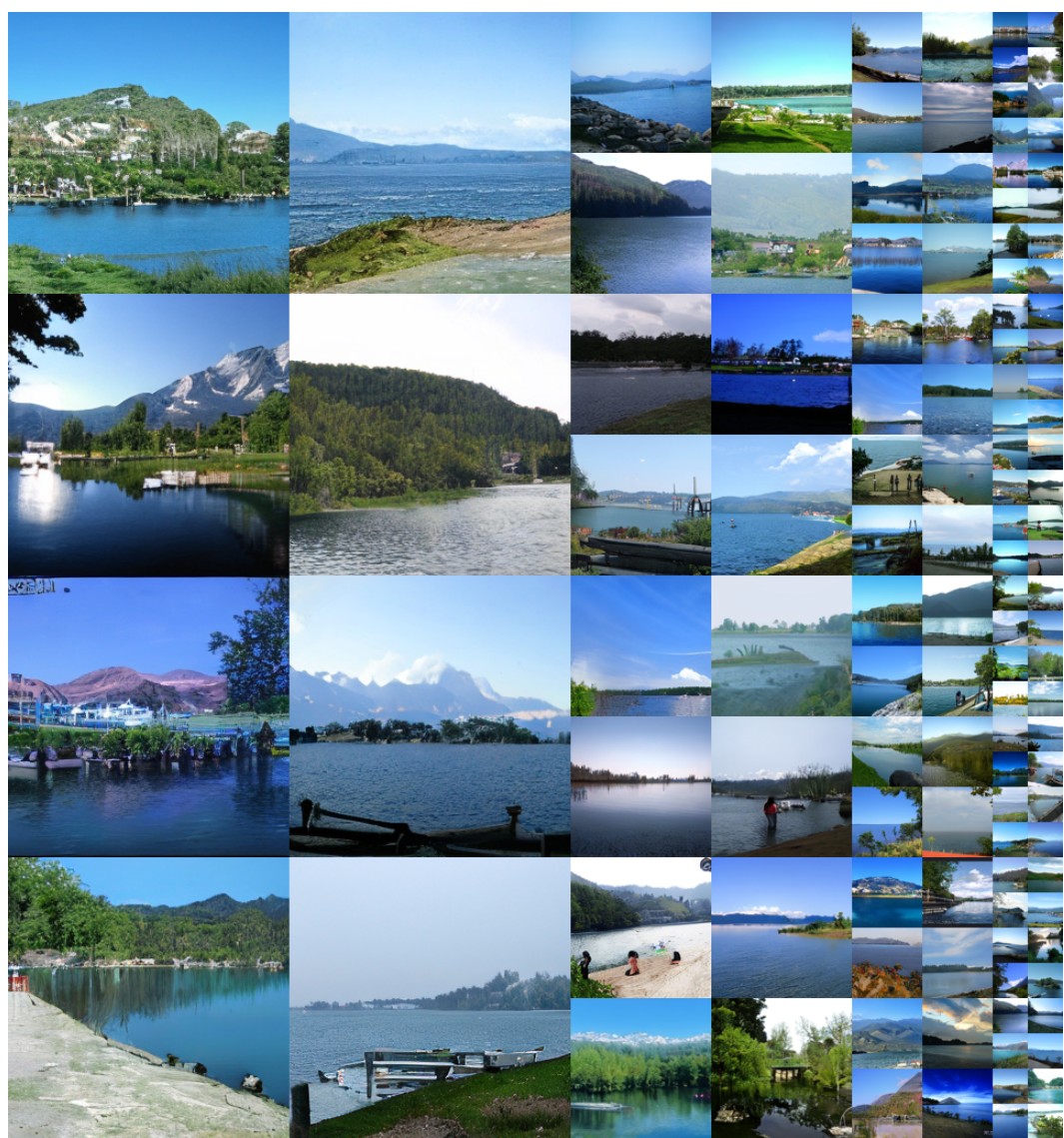

Figure 15: **Uncurated 256×256 DiT-L/2 samples.**
Classifier-free guidance scale = 2.0.
Class label = "lake shore" (975)

