# OpenReview forum: "Denoising Task Routing for Diffusion Models"
_ICLR.cc/2024/Conference — ICLR 2024 poster_

### Official Review · Reviewer_wtsR · 2023-10-30

**Soundness:** 4 excellent
**Presentation:** 3 good
**Contribution:** 3 good
**Rating:** 6
**Confidence:** 3

**Summary:**

In this paper, diffusion training is cast as multi-task learning, where each task corresponds to the denoising task at a specific timestep. The authors present Denoising Task Routing (DTR), a simple add-on strategy for existing diffusion model architectures to establish distinct information pathways for individual tasks within a single architecture by selectively activating subsets of channels in the model. Besides, the channel partitioning considers task affinity and task weights in diffusion models. Extensive experiments demonstrate the effectiveness and efficiency of the proposed method.

**Strengths:**

1. This paper proposes a simple add-on strategy for existing diffusion model architectures, which is simple yet effective, without introducing additional parameters, and contributes to accelerating convergence during training.
2. Extensive experiments demonstrate the effectiveness and efficiency of the proposed method.
3. The paper is well-written and easy to follow.

**Weaknesses:**

1. Some advanced routing methods [1, 2] improve the random routing by considering the inter-task relationship. Hence, it is better to discuss and compare the proposed method with them.

2. In Figure 9, the images generated by the baseline (the first row) look very strange and both R-TR and DTR methods alleviate it (the second and third rows). So why the random routing method can work well? In particular, in the fifth case/column, the image generated by R-TR looks better than the one generated by DTR. Why?

[1] Pascal et al. Maximum Roaming Multi-Task Learning. AAAI 2021.

[2] Ding et al. Mitigating Task Interference in Multi-Task Learning via Explicit Task Routing with Non-Learnable Primitives. CVPR 2023.

**Questions:**

Please address my concerns in the "Weaknesses" section.

---

> ### Author Response · Authors · 2023-11-18
> **Response to wtsR**
>
> Dear Reviewer wtsR,
>
> We appreciate the valuable feedback that you provide on our work. Your insights have been instrumental in refining our work and we are sincerely grateful for the opportunity to incorporate your constructive comments. In the following, we will thoroughly address each of the concerns raised and make the necessary revisions to improve the overall quality of our paper.
>
> ---
>
> ### **W1: Comparison with other routing methods**
> Thank you for providing us with further methods.
>
> **Max-roaming [1]**: We sincerely appreciate your recommendation. Comparing our method to max-roaming can further support the effectiveness of our method. We applied max-roaming to DiT-B/2 and trained it on the FFHQ dataset. The below table shows the comparative results between DTR and max-roaming.
>
> | Method      | FID      |
> |-------------|----------|
> | DTR         | **7.32** |
> | Max-Roaming | 39.90    |
>
> Consequently, their approach shows suboptimal performance due to its failure to incorporate the diffusion prior to task weight and task affinity. We suggest that this phenomenon is due to the detrimental effects of introducing randomness into the prior. Indeed, as illustrated by Go et al. [3], the application of random loss weighting has been observed to result in lower performance compared to the vanilla scenario.
>
> **ETR-NLP [2]**: utilizes slightly different routing with channel masking in DTR and TR and max-roaming [1]. Instead of task-specific masks in channel mask routing, task-specific layers (convolution layer) are employed for calculating representations in task-specific channels. Indeed, these task-specific layers require additional parameters, and they grow as the number of tasks increases, resulting in a huge cost in diffusion where there are a lot of denoising tasks (e.g. $T$ =1000). Meanwhile, our DTR does not require additional parameters for task routing. Also, their official code has not been released yet. In this regard, we did not conduct a direct comparison to ETR-NLP.
> Nevertheless, we tried to implement ETR-NLP in block-wise routing during the discussion phase, however, we failed to converge ETR-NLP despite trying various hyperparameters for training. We suspect that unaddressed elements in the scope of their paper might play a role in this outcome, and we will conduct experiments upon their code release.
>
> ---
> ### **W2: Quality of generated images**
> In some cases, DTR seems to perform worse than R-TR, while R-TR seems to outperform the baseline. However, the general trend observed is DTR >= Baseline >= R-TR. It is important to note that all qualitative results presented in the paper are uncurated samples, randomly selected without cherry-picking, which means that they may include instances where our approach encountered challenges or failures.
>
> ---
>
> ### **References**
>
> [1] Pascal et al. Maximum Roaming Multi-Task Learning. AAAI 2021.
>
> [2] Ding et al. Mitigating Task Interference in Multi-Task Learning via Explicit Task Routing with Non-Learnable Primitives. CVPR 2023.
>
> [3] Go et al. Addressing Negative Transfer in Diffusion Models. Neurips 2023.

---

> > ### Comment · Reviewer_wtsR · 2023-11-22
> >
> > Thank you very much for your response! It well addressed my concerns. Thus I will keep my initial rating towards acceptance.

---

> > > ### Author Response · Authors · 2023-11-22
> > > **Respond to Reviewer wtsR**
> > >
> > > We really appreciate the valuable and insightful comments provided by Reviewer wtsR, and we are glad to note that your concerns have been resolved. Thank you for taking the time to respond to the rebuttal.

---

### Official Review · Reviewer_VZ4w · 2023-11-01

**Soundness:** 3 good
**Presentation:** 3 good
**Contribution:** 3 good
**Rating:** 8
**Confidence:** 4

**Summary:**

This paper proposes Denoising Task Routing (DTR), a simple add-on strategy for existing diffusion model architectures to establish distinct information pathways for individual tasks within a single architecture by selectively activating subsets of channels in the model. The authors incorporate two prior knowledge aspects of diffusion-denoising tasks—task affinity and task weights—into the model architecture design to mitigate the negative transfer phenomenon. The paper provides empirical results on several image generation tasks and a qualitative analysis to validate the effectiveness of DTR.

**Strengths:**

* The paper effectively addresses the negative transfer phenomena by establishing task-specific pathways for multiple denoising tasks. The concept of integrating key prior knowledge in diffusion and task routing is well-presented and could potentially influence future work on architecture design in diffusion models.
* The implementation, although simple, is effective and yields significant performance gains on multiple benchmarks.
* The paper is structured well, making it easy to understand and follow.

**Weaknesses:**

* The empirical analysis could be more comprehensive in decoupling the contributions of task weights and task affinity. As I understand, the results in Figure 4 only ablate the significance of the synergy of the two priors. To study the direct contribution of **Task Weights**, it would be helpful to compare `DTR with random routing but task-dedicated allocation channels` with `Random Task Routing (R-TR)`. Similarly, to study the contribution of **Task Affinity**, a comparison between `DTR with task-unified allocation channels but sliding window channels` and `R-TR` would be useful.
* The paper does not adequately explain the sensitivity of DTR to different masking strategies. The authors should elaborate on why they chose Equation (4) as the masking strategy and discuss potential alternatives.

**Questions:**

* Would DTR achieve better performance by reducing the overlap channels at higher timesteps? Given the authors' assertion in Figure 6 that "at higher timesteps, the model primarily focuses on learning discriminative features that are relevant to specific timesteps, whereas at lower timesteps, the model tends to exhibit similar behavior across different timesteps," it appears that denoising tasks at higher timesteps have less correlation. Would assigning these tasks entirely distinct channels be beneficial?
* Is there a typographical error in Equation (4)? Should the first $t$ be replaced with $t-1$?

---

> ### Author Response · Authors · 2023-11-18
> **Response to Reviewer VZ4w (1/2)**
>
> Dear Reviewer VZ4w,
>
> Thank you for your insightful feedback on our paper. We are grateful for the opportunity to enhance our work through your constructive comments. Below, we will address all raised concerns by the reviewer and revise the paper accordingly.
>
> ---
>
> ### **W1: Studies for validating 1) task affinity and 2) task weights.**
>
> We agree that the contribution of both prior 1) task affinity and 2)  task weights should be analyzed in our work. In this regard, we would like to respectfully emphasize that the contributions of the performance for both priors are already validated in Figure 4.
>
> 1. *Task Affinity*: the random routing method (R-TR) routes the tasks without specific consideration of task affinity. As detailed in Section 4.2, the expected number of shared channels remains constant for any pairs of distinct denoising tasks. Contrarily, our Denoising Task Routing (DTR) utilizes a sliding window strategy, shifting activated channels by the timestep from 1 to $T$. This approach ensures that tasks at neighboring timesteps share similar channel sets, thereby embedding task affinity into the routing process. For instance, when $\alpha=1$ in Eq. (4), DTR shifts channels linearly, focusing solely on task affinity without considering task weights, as seen in Figure 2. The performance superiority of DTR ($\alpha=1$) over R-TR, as shown in Figure 4, serves as an empirical validation of the benefits of incorporating task affinity in DTR.
> 2. *Task Weights*: By adjusting $\alpha$, DTR also accounts for task weights alongside task affinity. When $\alpha > 1$, a higher number of dedicated channels are allocated to later timesteps, in line with our intention to assign greater weight to these timesteps. We compared the performance of DTR with $\alpha=1$ and DTR with $\alpha>1$, finding that the latter generally surpasses the former, as evidenced in Figure 4 and Table 4. This comparison substantiates the impact of considering task weights in our model.
>
> In summary, our results validate the efficacy of DTR in the context of both task affinity alone and in combination with task weights. We regret to inform you that we are currently unable to fully comprehend your queries regarding 'DTR with random routing but task-dedicated allocation channels with Random Task Routing (R-TR)' and 'DTR with task-unified allocation channels but sliding window channels and R-TR'. We kindly request further clarification on these points to provide a more thorough response. We eagerly anticipate a productive and enlightening discussion on these matters.
>
> ---
>
> ### **W2-1: Rationale for choosing Equation (4).**
> Thank you for pointing this out. As mentioned above in W1, our DTR firstly incorporates task affinity with activated channels as a sliding window according to timesteps, then modulates the shifting ratio of the sliding window across timesteps for consideration of task weights. In this regard, we implemented the idea as Eq. (4), However, we acknowledge that the conceptual explanation of Eq. (4) had been lacking in the manuscript. Through Eq. (4), each mask has $C_{\beta}$ sequential activated channels, while it is shifted as like a sliding window. In this situation, the available start index of activated channels is in {$0, \dots ,C − Cβ$}, and we quantized this start index according to timestep as $(\frac{t}{T})^{\alpha}$, enabling modulation of shifting ratio of the sliding window.
>
> ### **W2-2: Sensitivity of DTR masking strategies.**
> We would like to respectfully emphasize that we have validated the sensitivity of DTR masking strategies with regard to the hyperparameters $\alpha$ and $\beta$ in Figure 4 and Table 4.
>
> ### **W2-3: Discussion on potential alternatives.**
> Thank you for pointing this out. We agree that discussion on potential alternatives for masking strategies can promote future work by starting from our work. In our work, task-specific masks are not changed and optimized through training procedures. By utilizing well-known methods such as reinforcement learning and evolutionary algorithms, the masks can be more optimized than our DTR masks. Thanks again for pointing this out, we will add this discussion to our manuscripts.
>
> ---
>
> ### **Q1-1: Would DTR achieve better performance by reducing the overlap channels at higher timesteps?**
> We validated that reducing overlapped channels at higher timesteps improves the performance in Figure 4 and Table 4. By increasing $\alpha$ until adequate level, overlapped channels at higher timestep reduce. As shown in the results, the FID score is improved, achieving better performance.

---

> > ### Author Response · Authors · 2023-11-18
> > **Response to Reviewer VZ4w (2/2)**
> >
> > ### **Q1-2  Would assign these tasks entirely distinct channels be beneficial?**
> > In the typical configuration of the Diffusion model, where $T=1000$, employing distinct channels for each task becomes impractical. Assuming 16 channels per task, this would necessitate 16,000 channels, surpassing the 1024 channels of DiT-L, which we are currently handling. Therefore, we emphasize the impracticality of allocating distinct channels for each task.
> > Furthermore, claiming completely distinct channels implies disregarding the inter-task relationships in denoising tasks. As demonstrated by Go et al. [1], there are correlations between tasks at higher timesteps. We believe that leveraging these correlations and efficiently sharing channels can enhance parameter efficiency, as exemplified in our handling of the limited 1024 channels in DiT-L.
> >
> > ----
> >
> > ### **Q2: Typo**
> > We sincerely appreciate you pointing out typos. We will fix it.
> >
> >
> > ----
> >
> > ### **References**
> > [1] Go et al. Addressing Negative Transfer in Diffusion Models. Neurips 2023.

---

> > > ### Comment · Reviewer_VZ4w · 2023-11-23
> > > **Thanks for the response**
> > >
> > > The response for Week 1 thoroughly addresses my concerns, and I appreciate the detailed answers to the other questions, which have enhanced my understanding of the paper. Consequently, I will increase my rating to 8.

---

> > > > ### Author Response · Authors · 2023-11-23
> > > > **Thanks for the response**
> > > >
> > > > We would like to express our sincere gratitude for your valuable and insightful feedback. Your comments have significantly contributed to enhancing the quality of our work. We are pleased to know that the revisions and clarifications provided have effectively addressed your concerns.
> > > >
> > > > Thank you once again for your invaluable comments on our work.
> > > >
> > > > Sincerely,
> > > >
> > > > Authors

---

### Official Review · Reviewer_8VoW · 2023-11-02

**Soundness:** 3 good
**Presentation:** 4 excellent
**Contribution:** 2 fair
**Rating:** 8
**Confidence:** 5

**Summary:**

This paper introduces Denoising Task Routing (DTR), an add-on strategy for diffusion models that incorporates multi-task learning (MTL). The proposed channel masking strategy effectively boosts performance without introducing any extra parameters. The experiments demonstrate consistent improvement across evaluation protocols.

**Strengths:**

1. The proposed routing mask strategy is interesting as it leverages the task similarity between adjacent timesteps.

2. The experiment conducted in this study is comprehensive and demonstrates significant performance improvement.

**Weaknesses:**

1. The idea of considering diffusion models as multi-task learning has previously been proposed by Hang et al. (2023) and Go et al. (2023a). The proposed masking strategy in this work is a simple modification of TR (Strezoski et al., 2019). Its novelty is limited.

2. It lacks an ablation study to evaluate the necessity of the proposed masking strategy. Ding et al. (2023) propose to divide channels into shared channels and task-specific channels. Assigning each time-step cluster (Go et al., 2023a) to the respective task-specific channels can serve as an important baseline.

**Questions:**

See Weaknesses.

---

> ### Author Response · Authors · 2023-11-18
> **Response to Reviewer 8VoW**
>
> Dear Reviewer 8VoW,
>
> We are grateful for the insightful feedback you have provided on our work. Your perspectives have played a crucial role in refining our research and we sincerely appreciate the opportunity to incorporate your constructive comments. In the following, we address each of the concerns raised in detail and make the necessary revisions to improve the overall quality of our work.
>
> ---
>
> ### **W1: Limited novelty**
> Thank you for raising this important point, and we sincerely apologize for any confusion caused.
> While it is acknowledged that the diffusion model has been established in prior research as a multi-task learning framework for denoising tasks, it is noteworthy that the emphasis in these studies was primarily on optimization, with less consideration given to the architectural aspect. In the field of multi-task learning, it is widely accepted that both optimization and architecture play a pivotal role. We have shown that this architectural approach is the first point (in diffusion) and is also orthogonal to their methodology.
>
> In addition, Task Routing (TR) differs from our approach in several key respects. First, we have successfully eliminated the need for additional parameters when applying Task Routing (TR) to existing architectures. Second, our research emphasizes the importance of considering inter-task relationships and warns against a naive application that could lead to performance degradation. This underscores a critical message: performance improvements in the diffusion model are achievable by accounting for prior knowledge of denoising tasks.
>
> In light of these distinctions, we note that our contribution to task routing is significant and remains a valid addition when viewed through the lens of incorporating diffusion as a form of multi-task learning.
>
> ---
>
> ### **W2: Comparison with other potential baselines.**
>
> **Comparison with ETR-NLP (Ding et al. (2023) [2])**
>
> ETR-NLP divides channels into shared and task-specific channels, however, it utilizes slightly different routing with channel masking in DTR and TR and max-roaming [1]. Instead of task-specific masks in channel mask routing, task-specific layers (convolution layer) are employed for calculating representations in task-specific channels. Indeed, these task-specific layers require additional parameters, and they grow as the number of tasks increases, resulting in a huge cost in diffusion where there are a lot of denoising tasks (e.g. $T$=1000). Meanwhile, our DTR does not require additional parameters for task routing. Also, their official code has not been released yet. In this regard, we did not conduct a direct comparison to ETR-NLP.
>
> Nevertheless, we tried to implement ETR-NLP in block-wise routing during the discussion phase, however, we failed to converge ETR-NLP despite trying various hyperparameters for training. We suspect that unaddressed elements in the scope of their paper might play a role in this outcome, and we will conduct experiments upon their code release.
>
> **Comparison with the timestep cluster-based task-specific channels (Go et al, 2023a [3])**
>
> Thank you for suggesting a valuable recommendation. Comparing our DTR to task-specific channels which are allocated according to timestep clusters can show differences between routing with entire tasks and clustered tasks. We utilized timestep-based clustering [3] with ($k=8$) and assigned channel masks by regarding each cluster as one task and trained DiT on the FFHQ dataset. The below table shows the results, and our DTR dramatically outperforms the timestep cluster-based DTR, showing that routing entire tasks is more beneficial than routing clustered tasks. Thank you for improving our work, and we will add this to the manuscript.
>
>
> | Method                     | FID      |
> |----------------------------|----------|
> | DTR                        | **7.32** |
> | DTR w timestep-cluster [3] | 9.61     |
>
>
> ### **References**
> [1] Pascal et al. Maximum Roaming Multi-Task Learning. AAAI 2021.
>
> [2] Ding et al. Mitigating Task Interference in Multi-Task Learning via Explicit Task Routing with Non-Learnable Primitives. CVPR 2023.
>
> [3] Go et al. Addressing Negative Transfer in Diffusion Models. Neurips 2023.

---

> > ### Comment · Reviewer_8VoW · 2023-11-20
> >
> > Thank you so much for your response. There seems to be some misunderstanding in W2. Actually, what I meant is to first cluster the tasks and then assign them to shared channels and task-specific channels as in [2]. For instance, in a scenario with 8 task clusters and 32 channels, we can use 16 channels as shared channels, assign 2 task-specific channels to each task cluster. I wonder if the experiment result (i.e., DTR w timestep-cluster [3] in the table) include shared channels.

---

> ### Author Response · Authors · 2023-11-20
> **Response to the official comment by Reviewer 8VoW**
>
> Thank you for your clarifications regarding the concerns raised in your review. We appreciate your continued engagement with our work, and we're glad to have the opportunity to address the points you've mentioned. Below are our responses to your comments:
>
> ---
>
> ### **1. Misunderstanding in W2**.
> We appreciate the opportunity to address a potential misunderstanding highlighted by the reviewer in reference [2] regarding our ETR-NLP module. Concerning point [2], we wish to clarify a potential misunderstanding by the reviewer. In their ETR-NLP module referenced in [2], channels are categorized into two types: task-specific and shared. However, contrary to what the reviewer suggests, their task routing is conducted via a task-specific module, not through channel masking. Channel indices of both shared and task-specific channels are commonly shared across all tasks, with the shared channel representations processed by shared modules, while task-specific channel representations go through task-specific modules (such as convolutional layers).
>
> For instance, in a setup with 8 task clusters and 32 channels, 16 channels could be designated as shared, with the remaining 16 channels serving as task-specific across the 8 clusters. For each cluster, the shared layer processes the 16 shared channels, and concurrently, the corresponding task-specific layer handles the 16 task-specific channels. The outputs from these layers are subsequently concatenated. This example illustrates that the distinction between shared and task-specific channels is more nuanced than our task routing.
>
> Therefore, the example provided by the reviewer does not precisely align with the methodology used in [2]. However, to address your concern, we are currently conducting experiments on your suggested channel masking where half of the channels are designated as shared and the other half are equally divided among task-specific channels for each task. We will share the results once available.
>
> We hope this clarification provides a clearer understanding of our approach and resolves any misconceptions that may have arisen.
>
> ---
>
> ### **2. Whether shared channels are considered in experimental results in our response.**
>
> Regarding DTR w/ timestep-cluster [3] in our response, we initially grouped $T$ denoising tasks into 8 clusters. We then applied the standard DTR settings ($\alpha=4, \beta=0.8$) by treating each task group as a single task. Given that $\beta$ is set to 0.8, which is above 0.5, and the masks are allocated as a sliding window, certain channels function as shared channels, being commonly activated across all tasks.
> We note that $(2\beta - 1) \cdot C$ channels are shared across all tasks ($C$ indicates the total number of channels).

---

> > ### Comment · Reviewer_8VoW · 2023-11-23
> >
> > Thank you very much for your response. It has addressed all of my concerns. I will raise my score to 8.

---

> ### Author Response · Authors · 2023-11-21
> **Response to the official comment by Reviewer 8VoW - 2**
>
> ### **Experimental results regarding W2**
>
> We sincerely appreciate your patience while waiting for our results. In response to the reviewer's concern in W2, we configured half of the channels as shared across all denoising tasks, while the remaining channels were divided into $k=8$ segments, each activated for specific tasks. We denote this as Explicit Routing with Clustered Denoising Tasks (ERCDT) and illustrate the comparative results in the below table.
>
> | Method  | FID      |
> |---------|----------|
> | Vanilla | 10.99  |
> | ERCDT   |     10.13     |
> | DTR     | **7.32** |
>
>
> As shown in the results, our DTR dramatically outperforms ERCDT. This implies that such explicitly structured channel allocation may not be as effective as DTR. The primary reason for this disparity appears to be ERCDT's inability to sufficiently capture and reflect the nuanced, proximal relationships among denoising tasks due to its clustering approach. For instance, in ERCDT, the denoising task at $t=1$ is considered nearly equivalent to the task at $t=5$ or $t=100$ within the context of $k=8$ clusters, failing to recognize the higher affinity between tasks at closer time intervals. Furthermore, in the sequence of tasks, while $t=124$ and $t=125$ belong to the same cluster, timesteps $t=125$ and $t=126$ fall into different clusters. This quantized clustering doesn't effectively reflect one timestep difference between them. Consequently, this limitation hinders the performance of ERCDT compared to DTR, supporting the effectiveness of our DTR.
>
> However, ERCDT surpasses the performance of vanilla training. This result suggests that task routing incorporating the relationship of denoising tasks by task clustering boosts performance despite of its discreteness.
>
> We are profoundly grateful for your insightful suggestion. It has contributed significantly to our analysis, and we will ensure to include this valuable perspective in our paper.

---

> ### Author Response · Authors · 2023-11-23
> **Thanks for engaging discussion.**
>
> We sincerely appreciate the valuable and insightful comments provided by Reviewer 8VoW, which have been very helpful in improving our work. We are glad to hear that your concerns have been resolved. Thank you for your priceless effort to engage in discussion.
>
> Sincerely,
>
> Authors.

---

### Author Response · Authors · 2023-11-18
**Additional Results**

Dear reviewers,

We sincerely appreciate your dedicated efforts. Here, we show additional results to support the strengths of our method.

We trained DiT-L/2 + DTR with ANT-UW on more iterations, 2M, and achieved a **2.33** FID score in the ImageNet 256x256 dataset. The below table shows the comparison between vanilla DiT-XL and DiT-L/2 + DTR with ANT-UW. Our DTR with ANT-UW outperforms DiT-XL/2 trained on 2.35M iterations, despite using smaller parameters, Flops, and training iterations. Furthermore, our DTR with ANT-UW performs a similar level of DiT-XL trained on 7M iterations. We would like to emphasize that these results further support the effectiveness of our methodology.

| Method                   | FID      | Training iterations | # of parameters | Flops (G) |
|--------------------------|----------|---------------------|-----------------|-----------|
| DiT-XL/2-G               | **2.27** | 7M                  | 675M            | 118.64    |
| DiT-XL/2-G               | 2.55     | 2.35M               | 675M            | 118.64    |
| DiT-L/2-G + DTR w ANT-UW | _2.33_     | **2M**              | **458M**        | **80.71** |

---

### Author Response · Authors · 2023-11-22
**Summary of revision**

Dear Reviewers,

Thank you for your priceless efforts in reviewing our work. We have taken every suggestion seriously and have diligently worked to provide the best possible response. We encourage the reviewers to revisit the updated version of our paper, with revisions highlighted in **blue**.

---

### **The strengths emphasized by reviewers are:**

1. Our method is simple yet effective (VZ4w & wtsR).
2. The idea is interesting (8VoW).
3. The idea may have a large impact (VZ4w)
4. The paper is well-presented and easy to follow (VZ4w & wtsR).
5. Moreover, it leads to significant performance enhancement (8VoW &  VZ4w) supported by comprehensive experimentation (8VoW & wtsR)

In addition, we highlight that our method can simply _1) plug into any architecture_, requiring _2) no additional parameters_ and _3) significantly accelerate convergence speed_ with _4) just a few lines of code_.

---

### **To address the concerns articulated by the reviewers, we’ve undertaken revisions:**
1. **[8VoW] Novelty**: Our work introduces two significant innovations: a) Our Multi-Task Learning (MTL) approach, unlike prior task routing methods, requires no additional parameters or modules, making it highly efficient and adaptable to diffusion model architectures. b) We pioneer the integration of MTL architectural principles into diffusion models, a distinct departure from previous studies that focused on multi-object optimization.
2. **[8VoW, VZ4w, wtsR] Comparison with Other Alternatives**: We include a comprehensive comparison with alternative routing methods. The results, detailed in our experiments, demonstrate the superior performance of our Denoising Task Routing (DTR) when contrasted with established baselines.
3. **[VZ4w] Validation of Task Affinity and Task Weights, Sensitivity of DTR Masking Strategies**: For further clarification, we direct attention to Figure 4 and Table 4 in our paper.
4. **[VZ4w] Rationale for Choosing Equation (4)**: In our revised paper, we have elaborated on Equation (4), explaining how DTR effectively marries task affinity with a sliding window approach. This allows for dynamic adjustment of the shifting ratio across time steps, taking into account the varying task weights.
5. **[wtsR] Quality of Generated Images in Figure 9**: We've noted that some of our model-generated images may look less polished than others due to our practice of presenting unedited, randomly selected samples, reflecting our model's genuine performance.
6. **[VZ4w] Potential future work**: We have highlighted potential future works for promoting the impacts of our work.


We hope these updates will address the concerns raised in the initial reviews and provide a more complete understanding of our method. Thank you once again for your valuable feedback.

Sincerely,

Authors.

---

### Meta-Review · Area_Chair_Vuhd · 2023-12-15

**Metareview:**

This work presents Denoising Task Routing (DTR), an additional approach designed for integration with diffusion models, featuring the incorporation of multi-task learning (MTL). The suggested channel masking technique showcases its ability to improve performance without the introduction of any additional parameters. The experiments consistently illustrate notable performance improvements across various evaluation protocols.

All reviewers find the work well-motivated, well-presented, and the method is simple to implement, effective and versatile with convincing experiments.

**Justification For Why Not Higher Score:**

The contribution is a bit technical

**Justification For Why Not Lower Score:**

All reviewers agree that this is an interesting work with convincing results.

---

### Decision · Program_Chairs · 2024-01-16

Accept (poster)